# HAPLN1 potentiates peritoneal metastasis in pancreatic cancer

Lena Wiedmann [1,2,13], Francesca De Angelis Rigotti[1,3,13], Nuria Vaquero-Siguero[1], Elisa Donato [4,5], Elisa Espinet [4,5,6,7], Iris Moll[1], Elisenda Alsina-Sanchis [1,8], Hanibal Bohnenberger [9], Elena Fernandez-Florido[1], Ronja Mülfarth[1,2], Margherita Vacca[1], Jennifer Gerwing[1], Lena-Christin Conradi[10], Philipp Ströbel[9], Andreas Trumpp [4,5], Carolin Mogler [11], Andreas Fischer [1,8,12,14 ✉] & Juan Rodriguez-Vita [1,3,14 ✉]

Pancreatic ductal adenocarcinoma (PDAC) frequently metastasizes into the peritoneum, which contributes to poor prognosis. Metastatic spreading is promoted by cancer cell plasticity, yet its regulation by the microenvironment is incompletely understood. Here, we show that the presence of hyaluronan and proteoglycan link protein-1 (HAPLN1) in the extracellular matrix enhances tumor cell plasticity and PDAC metastasis. Bioinformatic analysis showed that HAPLN1 expression is enriched in the basal PDAC subtype and associated with worse overall patient survival. In a mouse model for peritoneal carcinomatosis, HAPLN1-induced immunomodulation favors a more permissive micro-environment, which accelerates the peritoneal spread of tumor cells. Mechanistically, HAPLN1, via upregulation of tumor necrosis factor receptor 2 (TNFR2), promotes TNF-mediated upregulation of Hyaluronan (HA) production, facilitating EMT, stemness, invasion and immunomodulation. Extra-cellular HAPLN1 modifies cancer cells and fibroblasts, rendering them more immunomodulatory. As such, we identify HAPLN1 as a prognostic marker and as a driver for peritoneal metastasis in PDAC.

Pancreatic ductal adenocarcinoma (PDAC) ranks among the most lethal cancer entities, with late diagnosis as key contributor to its poor survival rate, as most patients are detected with metastatic disease[1]. PDAC metastasis most commonly occurs in the liver[2]; however, many patients suffer from peritoneal carcinomatosis, which is a major, but understudied cause of morbidity and mortality of patients with no effective treatment options[3]. Indeed, at time of diagnosis around 9% of PDAC patients present with peritoneal metastasis, while the rate at death raises up to 25–50%[3]. Therefore, understanding the mechanisms of peritoneal dissemination is of critical clinical need to shed a light into new treatment options.

Metastatic tumor cells need to acquire certain features, which allow them to survive and grow out within a distant hostile micro-environment and to escape the immune response. All these

characteristics can be controlled by unlocking cellular plasticity, an event recently incorporated to the hallmarks of cancer[4]. Cellular plasticity is characterized by the ability of cells to convert between different (intermediate) cellular states by inducing epithelial-to-mesenchymal transition (EMT), as well as mesenchymal-to-epithelial transition (MET) and features of cancer cell stemness[5]. Plasticity not only makes cancer cells more prone for invasion and adaption to the microenvironment, but also protects them from apoptosis, immune attack and chemotherapy[6].

Despite the cancer cell intrinsic features, a key regulator of metastasis is the microenvironment that tumor cells are facing during their journey to the metastatic site[7]. The tumor microenvironment (TME) consists of several cellular and non-cellular components, including cancer-associated fibroblasts (CAFs), endothelial cells,

immune cells and extracellular matrix (ECM). In PDAC, the TME is strongly desmoplastic, with substantial accumulation of ECM components. Hyaluronic Acid (HA), one major component of the ECM, facilitates tumor progression and metastasis through promoting partial EMT, invasion, immunomodulation and therapy resistance[8]. CAFs are the main producers of ECM, however their additional role as immunomodulators has been increasingly recognized. By the expression of different cytokines, growth factors, and immunomodulatory molecules, CAFs impact on the recruitment, differentiation and polarization of innate and adaptive immunity[9].

When metastasizing into the peritoneum, disseminated tumor cells can often colonize milky spots of the omental fat pad, a metastatic niche with high nutrient levels and immune cells with more resolving, anti-inflammatory features (mostly resident macrophages and B cells)[10]. For instance, omental resident macrophages were proven crucial for metastatic progression and immunomodulation[11]. Thus, peritoneal and omental metastasis are driven by a combination of tumor cell intrinsic plasticity and niche.

When first colonizing the omentum, cancer cells progress to spread throughout the whole peritoneum. Suspended in the peritoneal cavity, tumor cells face different challenges, like anoikis, which can be prevented by cell cluster (spheroid) formation, maintenance of mesenchymal and stemness state (e.g. via STAT3 signaling) and/or survival signaling like PI3K, MEK-ERK, or TNF-TNFR2-NF-κB[3,12]. Additionally, tumor cells are facing tumoricidal immune cells. However, tumor-associated macrophages (TAMs), which are the predominant immune population in the diseased peritoneum, allow peritoneal metastasis from multiple tumor entities[11,13]. The phenotypic switch of resident tumoricidal peritoneal macrophages towards the TAM phenotype is affected by factors such as tumor-derived HA[14]. Thus, targeting HA is not only promising for localized, but also metastatic PDAC. However, clinical trials targeting HA with the hyaluronidase PEGPH20 failed in stage-3[15], therefore new treatment options are of utmost need.

Hyaluronan and proteoglycan link protein-1 (HAPLN1) is a HA and chondroitin sulfate proteoglycan (CSPG) crosslinker in the ECM, with so far poorly understood roles in cancer. Recently, CAF-derived HAPLN1 was found to fuel tumor cell invasion in gastric cancer and was associated with worse overall survival in pleural mesothelioma and drug resistance in multiple myeloma[16–18]. In contrast, HAPLN1 expression was associated with reduced disease progression in melanoma and colorectal cancer[19–21]. Additionally, in hepatocellular carcinoma HAPLN1 is expressed by tumor cells and associated with EMT[22]. Nevertheless, its functional role in PDAC remains elusive.

In this study we identified HAPLN1 as one of the most upregulated genes in PDAC compared to adjacent tissue. We define HAPLN1 as a mediator of peritoneal dissemination in PDAC, by inducing a highly plastic phenotype in cancer cells, which leads to a pro-tumoral metastatic niche.

## Results

### HAPLN1 is upregulated in PDAC tissue and associated with worse outcome

To unravel mediators of HA-mediated PDAC progression, we analyzed publicly available data sets of PDAC and adjacent tissue from patients (Cao et al. 2021, GSE62452,[23,24]). We performed gene set enrichment analysis (GSEA) using a gene set for HA-binding (Gene Ontology for "Hyaluronic Acid Binding") to compare its expression between tumorous and adjacent tissue (Fig. 1A, Supplementary Fig. 1A). Interestingly, HAPLN1 was the most enriched gene in one of the analyses, and within the leading edge in both data sets. The significant upregulation of HAPLN1 gene expression in tumorous tissue compared to healthy was confirmed using the TNMplot database[25] (Fig. 1B).

Further, we classified different tumorous tissue samples (Cao et al. 2021, GSE50827,[23,26]) as HAPLN1[high] or HAPLN1[low] based on mean

HAPLN1 expression level. By GSEA, we discovered a significant enrichment of the gene set for the basal PDAC subtype in the HAPLN1[high] samples (Fig. 1C; Supplementary Fig. 1B), while the gene set of classical PDAC subtype was enriched in HAPLN1[low] in one of the datasets (Fig. 1D, Supplementary Fig. 1C). The basal subtype is characterized by less differentiated cells and patients have a lower overall survival than those with the classical subtype[27]. Moreover, EMT-related gene sets were significantly enriched in HAPLN1[high] as observed by GSEA of the gene set "Hallmark: Epithelial-to-mesenchymal transition" (Fig. 1E, Supplementary Fig. 1D), reinforcing our hypothesis of HAPLN1 as a mediator of PDAC progression. Indeed, when addressing if HAPLN1 expression is associated with disease outcome, we found that overall survival of patients was significantly reduced in the HAPLN1[high] group, both when mean of HAPLN1 expression was assessed at the mRNA and protein level (Fig. 1F). All these data were further confirmed in patient data from the TCGA database (Supplementary Fig. 1E–G).

Next, we sought out to understand which cells within the tumor were producing HAPLN1. While immune and stromal cells expressed HAPLN1 mRNA in both normal and tumoral tissue, only the epithelial cells significantly increased HAPLN1 expression upon their transformation to tumor cells (Fig. 1G). Indeed, we observed that HAPLN1 protein expression in tumors of PDAC patients was localized in tumor cell areas (Fig. 1H). Given the high levels of HAPLN1 expression in stromal cells, we analyzed the patient clinical data provided in the study from Cao et al.[23], in order to exclude that the differences observed between HAPLN1[high] and HAPLN1[low] patients were a result of a different fibroblastic content. We observed that the stromal component of HAPLN1high and HAPLN1low tumors was comparable and even the epithelial content was increased in HAPLN1high patients (Supplementary Fig. 1H).

### HAPLN1 induces EMT and ECM remodeling in cultured PDAC cells

To understand the possible functional role of HAPLN1 in PDAC, we overexpressed HAPLN1 in a murine PDAC tumor cell line derived from KPC (Kras[G12D]; Trp53[R172H]; Elas-CreER[28]) mice. KPC cells have very low HAPLN1 expression under normal cell culture conditions (Supplementary Fig. 2A). KPC-HAPLN1 cells had a more mesenchymal appearance compared to KPC cells, which grew in islet-like cell associations (Supplementary Fig. 2B).

Since HAPLN1 acts as crosslinker between HA and proteoglycans in the ECM, we addressed if KPC-HAPLN1 cells showed changes in their HA production. Significantly more HA was detected in the supernatant of KPC-HAPLN1 cells (Fig. 2A). This was caused by an upregulation of HA-synthase 2 (HAS2), while the other synthases (Has1, Has3) remained unchanged (Fig. 2B, C). HAS2 is the main producer of high molecular weight (HMW)-HA and a known inducer of EMT[29]. Thus, we analyzed EMT traits by investigating changes of mRNA and protein levels of mesenchymal and epithelial markers. This confirmed a more mesenchymal state of KPC-HAPLN1 cells, with mesenchymal markers like TWIST1 and LRRC15 being upregulated, while the epithelial marker E-Cadherin was downregulated (Fig. 2D, E). Moreover, we observed increased motility, a feature of a more mesenchymal phenotype (Supplementary Fig. 2C, Supplementary Movie 1, 2), which was not mediated by an increased cell proliferation (Supplementary Fig. 2D, E).

Since HA does not only affect EMT, but also stemness[30], we investigated the effect of HAPLN1 on the expression of stemness-related genes Abca1, Plod2, Abcg1, c-Kit, Cd133 and Lrg5 (Fig. 2F, Supplementary Fig. 2C). The significant upregulation of all these markers in KPC-HAPLN1 cells prompted us to assess spheroid formation as a readout for stemness. KPC-HAPLN1 cells were capable of forming proper and round spheroids in 3D culture, while KPC cells only formed aggregate-like structures (Fig. 2G). Measuring the area and roundness as indicators of complete spheroids, we could confirm the optimal spheroid formation capacity of KPC-HAPLN1 cells as shown by a

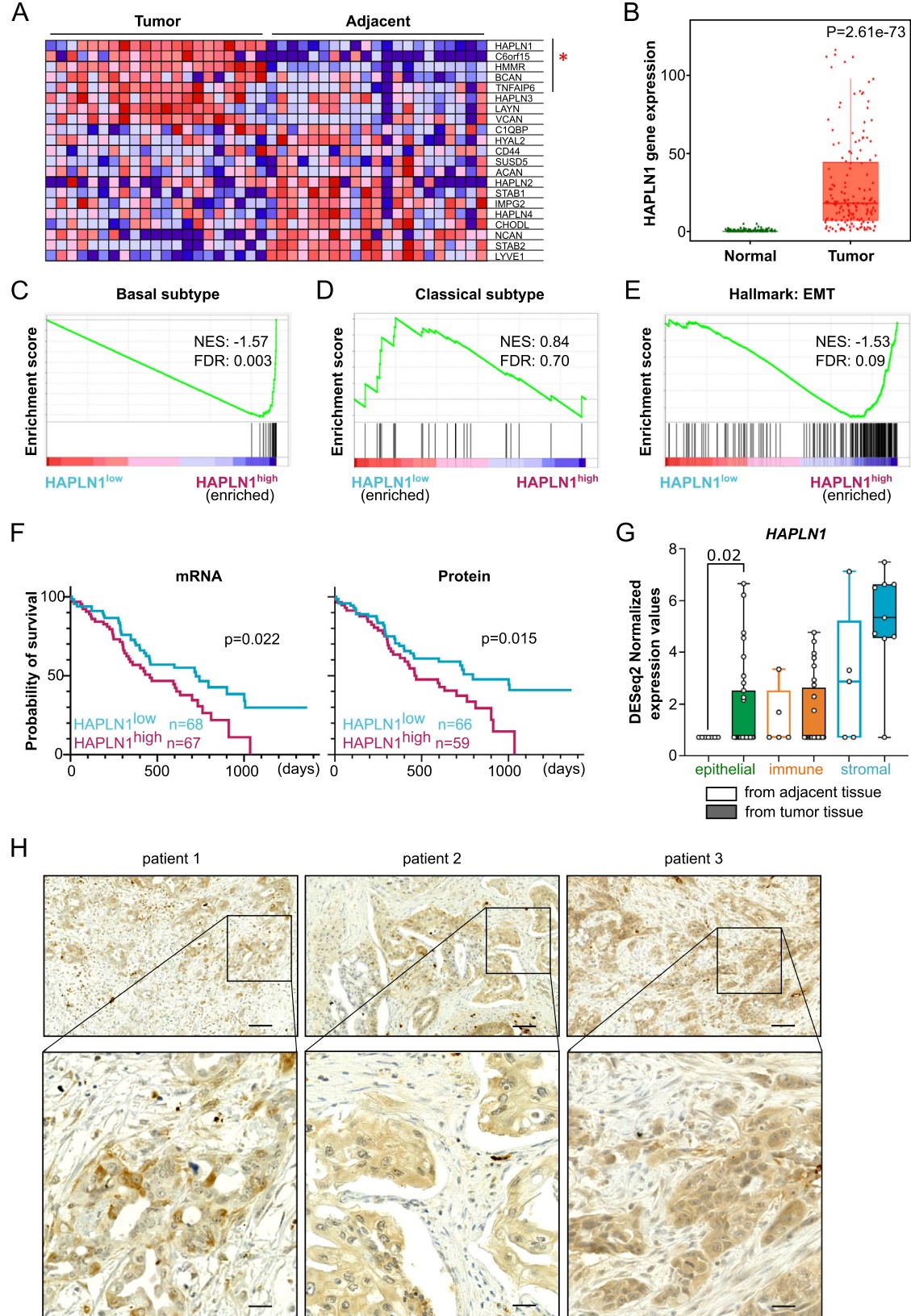

significant decrease of their area and a roundness of >0.9 (Fig. 2G). We reproduced this effect on the human PDAC cell line PANC1, in which HAPLN1 overexpression also improved spheroid formation (Supplementary Fig. 2G). Matching with the previous results, spheroids formed by KPC-HAPLN1 cells contained significantly more alive cells than spheroids formed by KPC cells (Fig. 2H).

The linker function of HAPLN1, crosslinking HA to proteoglycans (Fig. 2I), suggests that the improved spheroid formation could be mediated by changes in their ECM composition. Thus, we addressed gene expression of the HA synthase *Has2*, as well as of the proteoglycans aggrecan (*Acan*) and versican (*Vcan*) in spheroids. *Has2* and *Vcan* levels were significantly upregulated, while *Acan* levels were strongly

**Fig. 1 | HAPLN1 is upregulated in PDAC tissue and associated with worse disease outcome. A** GSEA of data set published in Cao et al.[23] (matched tumor and adjacent tissue samples. $n = 21$) on "Gene ontology for Hyaluronic Acid binding". Leading edge genes significantly deregulated are marked with red star. **B** HAPLN1 expression levels in normal adjacent or PDAC tumor tissue according to TNMplot, (Normal: Min = 0, Q1 = 0, Med = 0, Q3 = 0, Max = 5, Upper Whisker = 0, $n = 252$; Tumor: Min = 0, Q1 = 7, Med = 19, Q3 = 57, Max = 1960, Upper Whisker = 129, $n = 177$). Unpaired non-parametric Mann–Whitney two-tailed U test was applied. **C–E** GSEA on data set of Cao et al. (2021). Before analysis PDAC patients were divided in *HAPLN1* high/low according to mean *HAPLN1* expression. Gene sets of "Basal subtype" (**C**), "Classical subtype" (**D**) or "Hallmark of Epithelial-to-Mesenchymal transition" (**E**). $n = 140$. **F** Overall survival (OS) of PDAC patients stratified in two cohorts according to their mean HAPLN1 mRNA ($n = 135$) or protein expression levels ($n = 125$). For OS log-rank (Mantel–Cox) test was applied. **G** Human PDAC and adjacent tissue samples were sorted for epithelial/tumoral cells (EpCAM$^+$ CD45$^-$),

immune cells (EpCAM$^-$ CD45$^+$) and stromal cells (EpCAM$^-$ CD45$^-$). *HAPLN1* expression in cell types isolated from tumor by RNAseq; expression normalized to control tissue. Epithelial: Normal, Min = 0,7140, Q1 = 0,7140, Med = 0,7140, Q3 = 0,7140, Max = 0,7140, Upper Whisker = 0, $n = 7$; Tumor, Min = 0,7140, Q1 = 0,7140, Med = 0,7140, Q3 = 2,520, Max = 6,651, Upper Whisker = 1,746, $n = 31$. Immune: Normal, Min = 0,7140, Q1 = 0,7140, Med = 0,7140, Q3 = 2,511, Max = 3,343, Upper Whisker = 1,147, $n = 5$, Tumor, Min = 0,7140, Q1 = 0,7140, Med = 0,7140, Q3 = 2,623, Max = 4,765, Upper Whisker = 1,363, $n = 26$; Stromal: Normal, Min = 0,7140, Q1 = 0,7140, Med = 2,868, Q3 = 5,211, Max = 7,117, Upper Whisker = 2,622 $n = 5$; Tumor, Min = 0,7140, Q1 = 4,567, Med = 5,332, Q3 = 6,618, Max = 7,474, Upper Whisker = 2,001, $n = 9$. Dots represent biological replicates. Unpaired non-parametric Mann–Whitney two-tailed U test was applied. **H** HAPLN1 protein staining in human PDAC patient samples ($n = 20$) show staining in areas with tumoral cells. Three examples shown. Scale bar: 100 μm, zoom 25 μm.

---

reduced (Fig. 2J, K). In summary, these data demonstrate that HAPLN1 expression remodels the ECM and promotes EMT and stemness of pancreatic cancer cells.

## HAPLN1 promotes invasion

Partial EMT, motility, and stemness are signs of cellular plasticity. Another feature of high plasticity is invasion. To evaluate the role of HAPLN1 expression in invasion, we embedded tumor cell spheroids into Matrigel (Supplementary Fig. 3A) and evaluated their invasive potential. KPC cells were not able to invade the Matrigel. In contrast, KPC-HAPLN1 cells robustly invaded into Matrigel (Fig. 3A) and collagen gel (Fig. 3B). In line with this, also PANC1-HAPLN1 had increased invasive capacities into Matrigel compared to control (Fig. 3C), indicating that the effect of HAPLN1 is not restricted to KPC. Importantly, control cells formed round spheroids after the addition of Matrigel (Fig. 3A, C), confirming the importance of ECM in this process.

HA has been shown to act through CD44 activation[30]. Since HAPLN1 expression increases HAS2 and HA synthesis, we decided to evaluate the potential contribution of HA/CD44 to the increased migration promoted by HAPLN1. We found that neither HA hydrolysis by hyaluronidase, nor CD44 blockade by a neutralizing antibody reduced HAPLN1-induced invasion (Fig. 3D). However, HAS inhibition by 4-Methylumbelliferone (4-MU) completely abolished KPC-HAPLN1 invasion in Matrigel (Fig. 3D). This indicates that HAPLN1-induced invasion is dependent on HA synthesis but it is independent of either HA size or CD44 response to it.

## HAPLN1 expression affects also neighboring cells

Since HAPLN1 modified the composition of the ECM, we evaluated how a heterogenous tumor cell population would affect the tumor. For this, we mixed KPC and KPC-HAPLN1 cells at different ratios and assessed spheroid formation capacity (Fig. 3E). Here, already in a 4:1 (KPC:KPC-HAPLN1) ratio, spheroid formation capacity was significantly improved, measured by area and roundness (Fig. 3E). To understand whether HAPLN1 could also impact invasion of non HAPLN1-expressing cell, we stably labeled KPC and KPC-HAPLN1 with RFP or GFP. We co-cultured them as spheroids and embedded them into Matrigel. Interestingly, KPC cells invaded the Matrigel only when co-cultured with KPC-HAPLN1 cells (Fig. 3F). Additionally, we used adenoviral overexpression of GFP or mCherry to exclude an effect of lentiviral expression in the obtained results (Supplementary Fig. 3B). Next, in order to evaluate whether HAPLN1 presence in the ECM was enough to promote cell invasion we incubated KPC cells with recombinant HAPLN1 (rHAPLN1), exogenous HA or both. For this purpose, we used collagen for embedding to avoid any influence of possible HA (or other factors) in Matrigel. We found that either HA or HAPLN1 presence was enough to promote KPC invasion (Fig. 3G). These data indicate that HAPLN1 effects on KPC are cell extrinsic.

## HAPLN1 induces a highly plastic tumor cell state in vivo

To explore the effects of HAPLN1 on EMT, stemness and invasion in vivo, we used a model for peritoneal carcinomatosis, injecting RFP and luciferase-expressing KPC and KPC-HAPLN1 cells intraperitoneally (i.p.), to mimic an advanced tumor stage. This model is frequently used to study peritoneal dissemination and metastasis in other abdominal cancers, such as ovarian and gastric cancer, where tumor cells settle in the omental fat pads before spreading throughout the peritoneal cavity[11].

In vivo luminescence showed that KPC-HAPLN1 tumor-bearing mice depicted a more intense signal (Fig. 4A), which could not be attributable to an increased weight of the tumor masses present in the omentum (Supplementary Fig. 4A). When analyzing the cell composition of the tumor masses formed in the omentum by flow cytometry and immunofluorescence, we detected that immune cell distribution within the tissue was very heterogeneous (Supplementary Fig. 4B), with immune deserted tumor centers and high to moderate presence on the borders (Supplementary Fig. 4C), which prevented further quantification. Therefore, we next analyzed mRNA expression levels of bulk tumor samples to get a deeper understanding of the milieu. In KPC-HAPLN1 tumors, there was higher expression of the immunomodulatory markers *Cd274* (PD-L1) and MHC-II complex component *H2-Ab1*, as well of stemness markers *Cxcl12, Plod2* and *L1cam* compared to KPC control tumors (Supplementary Fig. 4D).

To assess plasticity-related processes, we sorted tumor cells (DAPI$^-$/CD45$^-$/CD31$^-$/RFP$^+$) by flow cytometry and investigated their expression profile by RNA sequencing. Principal component analysis (PCA) confirmed clustering of the two different groups, with almost all of the variation attributable to HAPLN1 expression (Supplementary Fig. 4E). When extracting significantly deregulated genes between KPC and KPC-HAPLN1 we found several markers of EMT (e.g. *Snai2, Twist1, Pdpn, Ocln, Krt8, Krt18*), stemness (e.g. *Cxcl12, Plod2, Osmr*), ECM remodeling (*Has1, Acan*) and immunomodulation (e.g. *H2-Ab1* (MHC-II), *Ccl2, Lcn2, Cxcl5, Cd47*) (Fig. 4B). We validated the upregulation of PDPN and MHC-II by flow cytometry. A high proportion of KPC-HAPLN1 cells expressed PDPN and MHC-II on their plasma membrane compared to the lack of staining on nearly all KPC cells (Supplementary Fig. 4F). The upregulation of PDPN in tumor cells was additionally confirmed by immunofluorescence staining (Fig. 4C). Gene Ontology (GO) analysis confirmed the link between HAPLN1 expression and "ameboidal-type cell migration", "positive regulation of cytokine production", "extracellular matrix organization", "regulation of Wnt signaling pathway" and "regulation of inflammatory response" enriched in KPC-HAPLN1 cells (Supplementary Fig. 4G). GSEA comparing KPC and KPC-HAPLN1 transcription profiles revealed that gene sets for "Hallmark: IL6-JAK-STAT3 signaling" and "Canonical Wnt signaling pathway" were significantly enriched in the KPC-HAPLN1 cells. STAT3 signaling was already shown to promote metastatic progression, including peritoneal

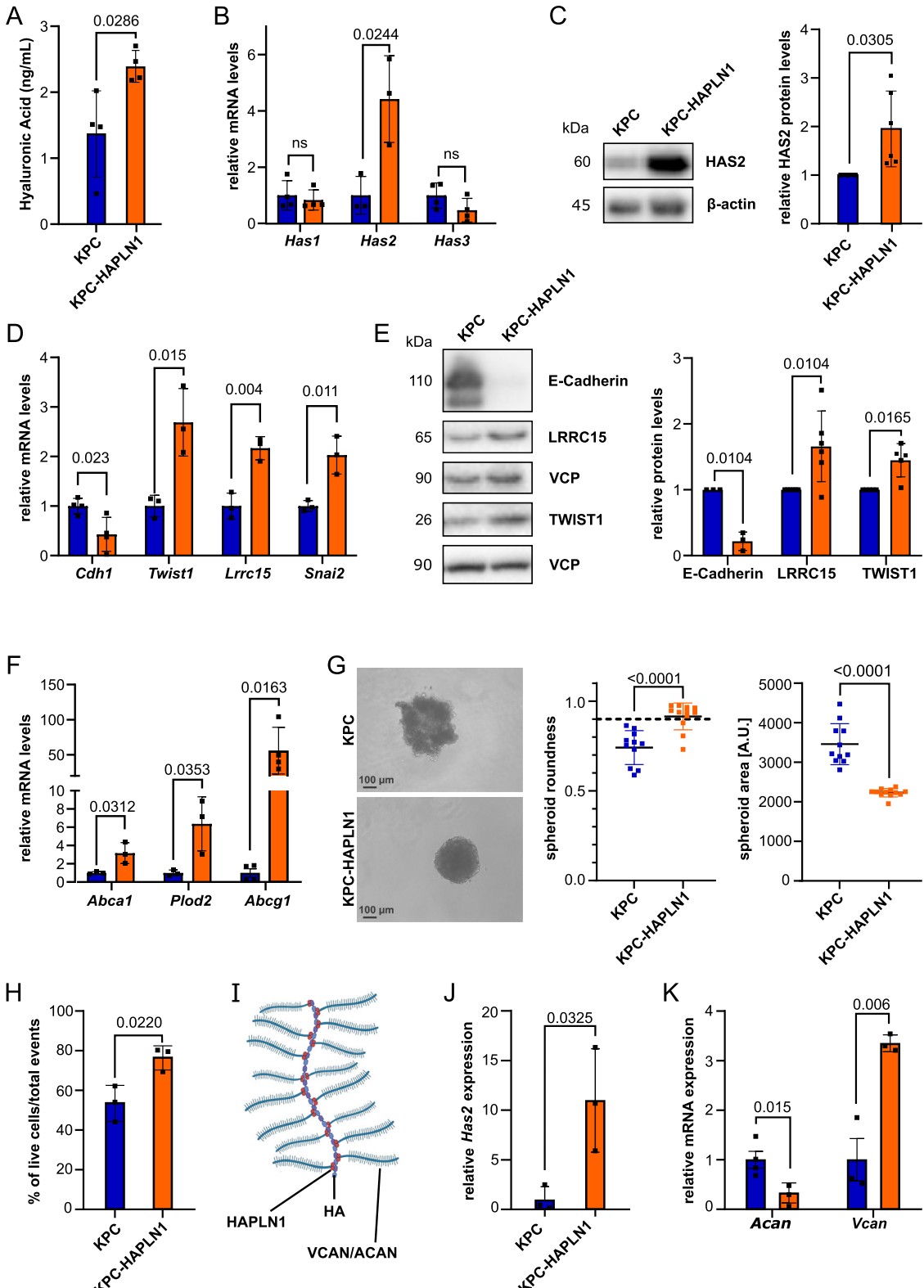

metastasis[31], while the enrichment of Wnt signaling corroborates a more stemness-promoting milieu[32] (Supplementary Fig. 4H). Immunofluorescence staining of pSTAT3 confirmed the increase in STAT3 activation in tumor cells (αSMA⁻/CD45⁻) from KPC-HAPLN1 tumor-bearing mice (Fig. 4D).

We decided to perform ingenuity pathway analysis (IPA) to confirm the role of IL6 in the promotion of peritoneal metastasis. However, we found TNF as the most significant common upstream regulator and

NFκB and TNF within the top 10 of genes with highest activation Z-score (Fig. 4E). In addition, when performing GSEA for the Hallmark for "TNFα signaling via NFκB" we confirmed its enrichment in KPC-HAPLN1 tumor cells (Fig. 4F). RNAseq analysis showed that TNFα receptor type 2 (*Tnfrsf1b*, TNFR2) significantly increased, which translated in the significant expression of several TNFα target genes, such as *Tnfaip3* or *Tnfaip6* (Fig. 4G). These data suggest that HAPLN1 presence in tumors promoted TNFα signaling by the upregulation of the TNFR2.

**Fig. 2 | HAPLN1 induces EMT and ECM remodeling in vitro. A** ELISA-like assay for Hyaluronan (HA) in the serum-free supernatant of KPC and KPC-HAPLN1 cells. $n = 4$. **B** qRT-PCR analysis of HA-synthase (Has) expression levels. HAS1 & HAS3, $n = 3$: HAS2, $n = 4$. **C** HAS2 protein levels evaluated by Western blot. Left panel: representative image, right panel: quantification by normalization to house keeper β-actin. $n = 6$. **D** mRNA expression of epithelial and mesenchymal markers comparing KPC and KPC-HAPLN1 cells. *Cdh1*, $n = 4$; rest, $n = 3$. **E** Western blot analysis of epithelial and mesenchymal markers on protein lysates. Left panel: representative image, right panel: quantification by normalization to house keeper VCP. E-Cadherin, $n = 3$; rest, $n = 6$. **F** mRNA expression of stemness markers by qRT-PCR. *Abcg1*,

$n = 4$; rest, $n = 3$. **G** Cells seeded in ultra-low attachment plates. Pictures taken after 48 h. Quantification of area as measure for proper spheroid formation. Scale bar: 100 μm. KPC, $n = 11$; KPC-HAPLN1, $n = 12$. **H** Flow cytometric analysis of DAPI⁻ events using spheroids formed for 48 h. $n = 3$. **I** Schematic overview of HAPLN1 function as crosslinker of HA and proteoglycans aggrecan (ACAN) or versican (VCAN). **J, K** Gene expression analysis of *Has2*, $n = 3$ (**J**) and HAPLN1-associated proteoglycans (**K**) in spheroids. *Vcan*, $n = 3$; *Acan*, $n = 4$. Plots are shown as mean ± SD. Data points represent independent biological replicates. For panels **A, B, D, F–J** unpaired two-tailed T test was applied, for panels **C, E** paired two-tailed T test.

## HAPLN1 induced plasticity is mediated by TNFR2 signaling

In order to investigate whether TNFR2 regulation was a direct action of HAPLN1, we analyzed the expression of these receptors in KPC-HAPLN1 vs. KPC. We found that the differential regulation of the TNF receptors seen in vivo was reproduced in vitro in the absence of other cells (Fig. 5A), indicating that HAPLN1 directly induces *Tnfrsf1b* (TNFR2) expression. When analyzing whether this receptor regulation could have an impact on TNFα responses, we found that HAPLN1 significantly increased p65 phosphorylation (Fig. 5B) and the expression of *Tnfaip6* upon TNFα stimulation (Fig. 5C, Supplementary Fig. 5A). Moreover, we observed that stimulating cells with TNFα increased invasion (Fig. 5D, Supplementary Fig. 5B), while inhibiting TNFα abolished HAPLN1-induced invasion (Fig. 5E, Supplementary Fig. 5B). We previously showed that HAPLN1-induced invasion was mediated by HA in a CD44-independent manner (Fig. 3D). In keeping with this, TNFα-induced invasion was not mediated by CD44, but dependent on HA synthesis (Fig. 5F). Interestingly, HAPLN1 significantly increased TNFα-dependent expression of *Has2* but not of *Cd44* (Fig. 5G), suggesting that TNFR1 could be responsible for *Cd44* expression, while *Has2* could be regulated through TNFR2. TNFRs transduce their signals through different components to induce different responses. While TNFR1 induces proliferation and is mediated in part by ERK activation, TNFR2 induces migration and its signal is transduced by PI3K, FAK and STAT3[33–35]. Interestingly, we found that invasion was dependent on PI3K, FAK and STAT3 signaling, while ERK inhibition increased even more the invasive potential of cells (Supplementary Fig. 5C). Interestingly, ERK inhibition increased invasion capacity also in cells that do not overexpressed HAPLN1, corroborating that ERK activation negatively regulates invasion (Supplementary Fig. 5D). In keeping with this, we found that KPC-HAPLN1 cells had FAK and STAT3 signaling activated as shown by their increased phosphorylation (Supplementary Fig. 5E). All together we show that HAPLN1 induces cell invasion by an increased sensitivity to TNFα via TNFR2 upregulation.

## HAPLN1 modifies the immune microenvironment

Flow cytometry of tumor masses in the omentum showed an increased presence of CAFs (PDPN⁺/CD31⁻/CD45⁻/RFP⁻) in KPC-HAPLN1 tumor-bearing mice (Supplementary Fig. 4B), which was confirmed by immunofluorescence analysis (Fig. 6A). As observed with flow cytometry (Supplementary Fig. 4B), endothelial cell (CD31⁺) presence was unchanged between both groups (Fig. 6A).

Given the strong presence of CAFs in these tumors, we decided to investigate also their expression profile by RNAseq. We sorted fibroblasts (DAPI⁻/CD45⁻/CD31⁻/RFP⁻/PDPN⁺) from KPC and KPC-HAPLN1 tumors by flow cytometry. Interestingly, the expression profile of CAFs clustered depending on the tumors from which they were obtained (Supplementary Fig. 6A), indicating that HAPLN1 presence in the ECM is sufficient to deploy a distinguishable phenotype also in stromal cells. We found that fibroblasts from KPC-HAPLN1 tumors significantly upregulated immunomodulatory and inflammation-related genes like *Lif, Il6, Il33, Il34, Ly6a, Ly6c1, Mx1* and *Has1* compared to fibroblasts from KPC tumors. In contrast, fibroblasts from KPC tumors expressed more intensely genes related to ECM-producing fibroblasts like *Col1a1, Col1a2, Fn1, Pdgfrb, Myo1b,* and *Heyl* (Fig. 6B). Interestingly, seeding

fibroblasts (GRX cells) on top of matrix produced by KPC or KPC-HAPLN1 cells was sufficient to modify their expression profile. *Col1a2* expression was higher in those fibroblasts seeded on KPC matrix while those on KPC-HAPLN1 matrix expressed more *Lif* (Supplementary Fig. 6B).

Since both, tumor cells and CAFs from KPC-HAPLN1 tumors showed changes in markers of inflammation and immunomodulation, we analyzed blood from tumor-bearing mice to evaluate whether the inflammatory milieu could even be detected systemically. We noticed a significant induction of IFNγ and inhibin-A (INHBA) protein levels in serum of KPC-bearing mice compared to control, tumor-free mice. This increase was abolished nearly to baseline in KPC-HAPLN1 tumor-bearing mice (Fig. 6C). Moreover, the immune checkpoint molecule PD-L1 was upregulated in KPC-HAPLN1 compared to control and KPC-injected mice, although this difference did not reach statistical significance (Fig. 6C). In addition, type-2 inflammatory mediator IL33 was only upregulated in KPC-HAPLN1 tumor bearing mice (Fig. 6C). These data support the hypothesis that HAPLN1 promotes an anti-inflammatory state. To understand the impact of these systemic changes on the peritoneal cavity, we analyzed the peritoneal lavage (PL) of tumor-bearing mice. Firstly, we detected a strong increase in the number of cells present in the peritoneal cavity of KPC-HAPLN1 compared to control KPC-bearing mice (Fig. 6D). When analyzing the immune compartment, we found a much higher percentage of B lymphocytes and macrophages in the PL of KPC-HAPLN1 compared to KPC tumor-bearing mice, while the percentages of monocytes and neutrophils were significantly lower (Fig. 6E, Supplementary Fig. 6C). We also observed a drastic reduction in eosinophil percentages, while helper, effector and regulatory T cell populations were unchanged (Supplementary Fig. 6C, D).

Tumor-associated macrophages (TAMs) are one of the most important mediators of peritoneal immunity and their proportions are directly associated with worse progression in different cancer entities[36,37]. To understand if TAMs in KPC-HAPLN1 tumor-bearing mice could be contributing to tumor progression as well, we isolated them and performed qRT-PCR analysis. This revealed reduced expression levels of pro-inflammatory markers like *Nos2* (iNOS) or *Inhba* in macrophages derived from KPC-HAPLN1 mice (Fig. 6F). In addition, when stimulating bone marrow-derived macrophages (BMDMs) with cancer cell conditioned medium, we detected significantly higher levels of the anti-inflammatory macrophage marker *Arg1* (Supplementary Fig. 6E) when KPC cells expressed HAPLN1, indicating that HAPLN1 in ECM promotes TAM education.

## HAPLN1 facilitates peritoneal colonization by tumor cells

The data presented so far indicate that HAPLN1 expression in PDAC cells induces a more aggressive, highly plastic state and a more tumor-permissive niche within the peritoneal cavity. For this reason, we decided to further investigate peritoneal dissemination of tumor cells.

We detected increased luminescence in peritoneal lavage from KPC-HAPLN1 compared to that from KPC tumor-bearing mice, which directly correlates with increased presence of tumor cells in peritoneal fluid (Fig. 7A). As mentioned before, this increased luminescence in the peritoneal lavage was not attributable to an increased size of the tumor

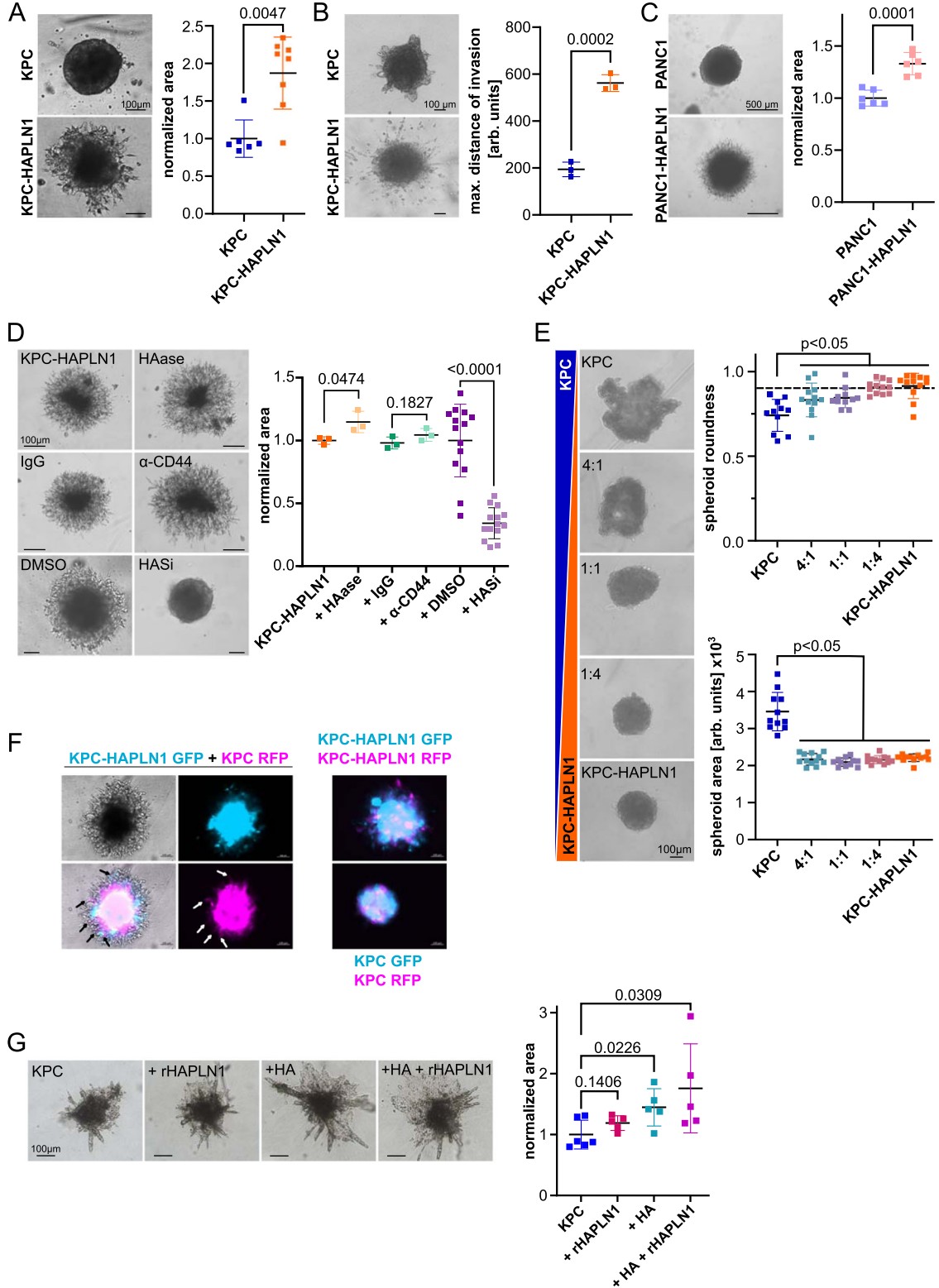

masses present in the omentum (Supplementary Fig. 4A). Flow cytometry of PL confirmed that there were more detached RFP+ tumor cells in KPC-HAPLN1 compared to control KPC tumor-bearing mice (Fig. 7B). These data indicate that KPC-HAPLN1 tumor cells are better equipped to invade and survive in the peritoneal cavity.

RNAseq analysis of ascitic tumor cells revealed some interesting traits acquired to survive in the peritoneal cavity. We isolated KPC-HAPLN1 tumor cells from the peritoneal lavage by cell sorting (DAPI-/CD45-/RFP+) and performed RNA sequencing.

Omental and ascitic tumor cells clustered in two separated groups when performing principal component analysis (PCA) (Supplementary Fig. 7A). Among the genes expressed significantly higher in the ascitic population, we found stemness-associated genes like *Cxcl12* and *Abca1*, peritoneal metastasis-associated genes, e.g. *Ctnnd1*, *Itga4*, *Limk1*, and immunomodulatory genes like *Cd36* (Fig. 7C). It is important to remark that the expression of apoptosis-inhibitors *Birc2* and *Bcl2l1* and immunomodulatory mediators *Pdpn*, *Cd47*, *Osmr* and *Muc16* increased progressively

**Fig. 3 | HAPLN1 fuels invasion and acts in a paracrine manner. A** Invasion of KPC or KPC-HAPLN1 cells 48 h after embedding of the spheroids into Matrigel. The occupied area of spheroids embedded into Matrigel is quantified as a readout of invasion (KPC: $n = 6$, KPC-HAPLN1: $n = 8$) and representative images are shown. Scale bar: 100 μm. **B** Invasion of KPC and KPC-HAPLN1 cells into collagen matrix. Distance of the three most invaded cells to spheroid was measured and averaged per sample. $n = 3$. Scale bar: 100 μm. **C** Invasion of PANC1 and PANC1-HAPLN1 cells into Matrigel 48 h after embedding. $n = 6$. Scale bar: 500 μm. **D** KPC-HAPLN1 cells embedded into Matrigel. Pictures taken 48 h after stimulation and embedding. Treatment with hyaluronidase (HAase, 200 mg/ml, $n = 3$), IgG control antibody (1 μg/ml, $n = 3$), anti-CD44 blocking antibody (1 μg/ml, $n = 3$), or the HAS inhibitor

4-methylumbelliferrone (HASi, 500 μM) or DMSO ($n = 14$). Scale bar: 100 μm. **E** KPC and KPC-HAPLN1 cells were mixed at different ratios. Images are representative of each condition. Scale bar: 100 μm. Spheroid roundness and area are quantified by Fiji software. KPC and 1:1, $n = 11$; rest, $n = 12$. **F** KPC and KPC-HAPLN1 cells were labeled with GFP or RFP via lentiviral vectors. Invasion was assessed after 48 h. Representative images of 3 independent experiments. Scale bar: 100 μm. **G** Invasion of KPC cells into collagen matrix 96 h after embedding and treating with exogenous HA (10 μg/ml) and/or rHAPLN1 (80 ng/ml). KPC, $n = 6$; rest, $n = 5$. Scale bar: 100 μm. Graphs represent mean ± SD. Each data point represents a biological replicate. Unpaired two-tailed T test was used for analysis.

from KPC to ascitic KPC-HAPLN1 (Fig. 7D). On the other hand, the epithelial cell markers *Krt18* and *Ocln* were highly expressed in ascitic vs. omental tumor cells (Fig. 7C). This suggests the acquisition of a more epithelial state, a process which was shown to be important for the attachment to and colonization of secondary organs[38]. Interestingly, *Cd47* is a "don't eat me" signal, which indicates that tumor cells would avoid being phagocytosed by immune cells. We attempted to analyze this by evaluating the number of immune cells (mostly macrophages) that had phagocytosed tumor cells (CD45$^+$RFP$^+$) in the peritoneal lavage. There was a significantly higher percentage of immune cells that phagocytosed tumor cells in KPC injected mice than in those injected with KPC-HAPLN1 (Supplementary Fig. 7B). As expected, when performing GO term analysis, almost all terms enriched in disseminated tumor cells were associated with immunity, inflammation and migration (Fig. 7E), suggesting that disseminated KPC-HAPLN1 cells could have cell-intrinsic features to modulate immune response and shape the microenvironment they are facing.

In order to validate these results, we analyzed the above-mentioned PDAC patient datasets classified as *HAPLN1*$^{high}$ and *HAPLN1*$^{low}$ and performed GSEA using a gene set of differentially expressed genes in peritoneal metastasis of patients with gastric cancer[39]. In both data sets (Cao et al 2021, GSE50827), the peritoneal metastasis gene set was enriched in *HAPLN1*$^{high}$ patients, reinforcing the hypothesis that HAPLN1 increases the potential of PDAC tumors to develop peritoneal carcinomatosis (Fig. 7F, Supplementary Fig. 7C).

Overall, we conclude that HAPLN1 expression in PDAC tumors promotes a highly plastic phenotype that facilitates invasion and colonization of the peritoneum, among others by generating a pro-tumoral immune microenvironment (Fig. 8).

## Discussion

The peritoneum is the second most common metastatic site in PDAC, with peritoneal metastasis contributing significantly to the devastating prognosis of patients. Our work identifies the extracellular protein HAPLN1 as a driver of tumor cell plasticity, promoting EMT, stemness, invasion and immunosuppression in a paracrine manner.

Cellular plasticity, as a key feature of tumor progression, metastasis and therapy resistance, is part of the latest list of "cancer hallmarks"[4]. ECM composition and remodeling are important factors influencing cellular plasticity. Here we addressed the function of the ECM modifier HAPLN1, a linker protein for HA and proteoglycans. The reasons to investigate HAPLN1 functions were based on the finding that HAPLN1 was one of the most upregulated HA-related genes in PDAC patients and on the fact that HA is a critical component of the tumor microenvironment, modifying EMT and invasion capacity of cancer cells.

Our study showed an impact of HAPLN1 on various different cell types in vitro and in vivo. In KPC cells, HAPLN1 expression resulted in the upregulation of EMT, stemness, invasion and peritoneal colonization. On top of the upregulation of several mesenchymal transcription factors like *Twist1* or *Snai2*, we found PDPN upregulated in KPC-

HAPLN1 cells. We showed that HAPLN1 presence in the extracellular matrix was already sufficient to trigger all these cellular changes, facilitating cell invasion. Interestingly, some of these features could be mimicked by stimulating with TNFα and inhibition of TNFα reverted the effects of HAPLN1, indicating that TNFα is responsible for the tumor cell plasticity observed in the presence of HAPLN1. Our data also suggest that TNFα promoted peritoneal carcinomatosis in vivo. This opens the possibility to employ TNFα inhibitors to prevent peritoneal carcinomatosis in PDAC patients. The common use of these inhibitors to treat chronic inflammatory diseases demonstrates the security of such treatments[40].

Further, our data indicate that tumor cells with high HAPLN1 expression become more independent of CAFs as matrix producers. These cells are more adapted to survive in suspension in the peritoneal cavity, after detachment from the tumor and its stroma. Interestingly, it was already shown that also circulating tumor cells (CTCs) in pancreatic cancer expressed high levels of ECM proteins compared to their counterparts in the solid tumor, which might protect them from immune cells, similar to platelets, and ease their survival by the ability to form clusters[41].

Regarding the impact on the immune system, CAFs in HAPLN1-rich environment shifted from ECM producers in KPC tumors to more immunomodulatory, which combined with the altered transcriptome of tumor cells could be the cause for the changes in the immune compartment within the peritoneal cavity. Both, CAFs and tumor cells have been shown to act as immune modulators in cancer, promoting tumor progression by creating favorable niches[42,43]. For instance, by the expression of CD47, a "don't eat me" signal, tumor cells evade phagocytosis by macrophages[44]. Interestingly, we found that higher levels of *Cd47* correlated with reduced phagocytosis of tumor cells by immune cells. We also found several modulators of inflammation upregulated in KPC-HAPLN1 cells, e.g. *Ccl2*, *Lcn2* or *Cxcl5*, which have been shown to regulate immune cell infiltration, contributing to disease progression[45–47].

In PDAC, several immune cell types have been described to promote different stages of tumor progression, including B cells[48], TAMs[49] and others. Data of immune cells in the peritoneal cavity during PDAC progression is however sparse and inconclusive. In other tumor entities colonizing the peritoneum, macrophages were found to be a key promoter of dissemination into the peritoneum and survival of detached cells[14,50]. In KPC-HAPLN1 tumor-bearing mice the presence of macrophages was significantly increased, with a more alternative, anti-inflammatory expression profile. This could form the base of an immune suppressive environment that would enable the immune evasion of disseminated KPC-HAPLN1 tumor cells in the peritoneal cavity, promoting tumor progression. Indeed, LIF, which was upregulated in fibroblasts in KPC-HAPLN1 tumors, has been shown to potentiate anti-inflammatory responses in TAMs to prevent CD8$^+$ T-cell infiltration[51]. The impact of the other immune cell types that were differentially recruited could not be unraveled in detail and needs further investigation.

Assessing HAPLN1 levels in PDAC patients could have diagnostic significance. It could serve to identify patients with a predisposition for

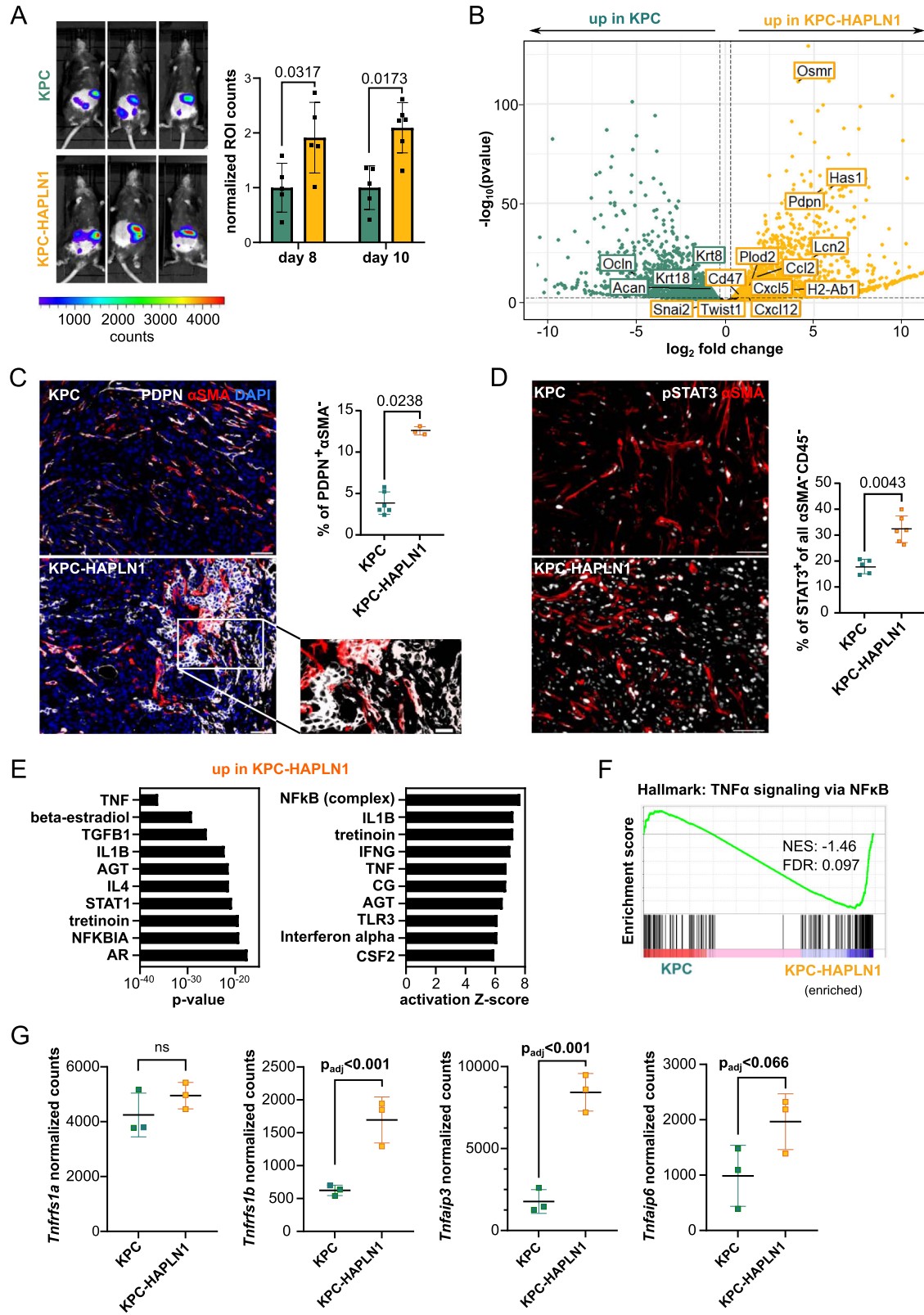

peritoneal metastasis and open the possibility for early changes in treatment strategies. Since most patients that acquire peritoneal metastasis throughout disease progression acquire it after primary tumor detection (9% of patients with peritoneal metastasis at time of detection vs. 25–50% at time of death[3]) the classification by HAPLN1 could reduce the acquisition of peritoneal metastasis, thereby improving patients' outcome. Nevertheless, more investigation is

needed to gain better mechanistic and overall understanding of HAPLN1 action within the tumor microenvironment. In this sense, it is important to notice that one of the genes associated with HAPLN1 expression was MUC16, also known as cancer antigen 125 (CA-125), which has been associated to PDAC progression[52].

Taken together, we could demonstrate that HAPLN1 expression in tumor cells modulates TNFα signaling to promote a highly plastic

**Fig. 4 | HAPLN1 induces a highly plastic tumor cell state in vivo. A** In vivo luminescence was measured at day 8 and 10 after tumor cell injection and normalized to the control. Three representative images per group are shown. KPC $n = 5$; KPC-HAPLN1 $n = 6$. **B** Volcano plot obtained from RNA sequencing on tumor cells that were isolated from KPC or KPC-HAPLN1 solid tumors. Each dot represents one gene, with green dots upregulated in KPC and yellow dots upregulated in KPC-HAPLN1 cells. Some genes of interest were labeled. $n = 3$. **C** Podoplanin (PDPN)/ αSMA staining on tumors. Representative images and quantification of PDPN⁺ αSMA⁻ cells are shown. Scale bar: 50 μm, zoom: 20 μm. KPC $n = 6$; KPC-HAPLN1 $n = 3$. **D** pSTAT3/αSMA staining on tumors. Representative images and

quantification of STAT3⁺ tumor cells are shown. KPC $n = 5$; KPC-HAPLN1 $n = 6$. **E** Top 10 upstream regulators assessed by ingenuity pathway analysis (IPA) of sorted tumor cells from mouse tumors sorted by $p$ value or activation Z score. **F** GSEA of "HALLMARK: TNFα signaling via NFκB" on sorted tumor cells. **G** Normalized read counts of TNFα-related genes in sorted tumor cells from mouse tumors. $n = 3$. Graphs represent mean ± SD. Data points indicate independent biological replicates. For panels **A**, **C**, **D** non-parametric two-tailed Mann–Whitney U test was used. For panel **B**, the package DESeq2 in R studio was used. In this package, the Wald test is used for hypothesis testing when comparing two groups. Correction was performed by multiple testing using the Benjamini–Hochberg (BH) method.

phenotype that facilitates invasion and colonization of the peritoneum, among others by creation of a pro-tumoral immune microenvironment.

## Methods

All mouse experiments were approved by the local authorities (RP Karlsruhe and DKFZ) and carried out following their legal requirements. Approval for using the resection samples was obtained from the institutional ethics committee (#24-4-20). All procedures were designed and performed in compliance with the Declaration of Helsinki and all institutional, state and federal guidelines.

### Patient samples

Pancreatic cancer patient histology samples were obtained from the Institute of Pathology of the University Medical Center Goettingen.

### Patient expression data of human epithelial, immune and stromal PDAC cells

Transcriptomic data of PDAC epithelial, immune, and stromal cells were obtained from PDAC specimens of patients who received partial pancreatoduodenectomy at the Department of General, Visceral and Transplantation Surgery, University of Heidelberg. Patients were part of the HIPO-project. The study was approved by the ethical committee of the University of Heidelberg (case number S-206/2011 and EPZ-Biobank Ethic Vote #301/2001) and conducted in accordance with the Helsinki Declaration; written informed consent was obtained from all patients. Epithelial, immune, and cancer-associated fibroblast populations were isolated via were isolated by fluorescence activated cell sorting using FITC anti-human CD326 (EpCAM, 1:11, Clone: AC128, Miltenyi Biotec) and VioBlue anti-human CD45 (1:11, Clone: 5B1, Miltenyi Biotec) after cell death exclusion using propidum iodide. RNA extraction, library preparation and RNA sequencing were performed as described elsewhere[53].

### qRT-PCR

For transcriptomic analysis, cell lines were lysed, and RNA was isolated according to manufacturer's protocol of the innuPREP RNA Mini Kit (Fisher Scientific, 10489573). Bulk tumor samples were homogenized with metal beads and RNA isolated with help of Trizol reagent (Life Technologies, 15596026) and isopropanol precipitation. RNA of sorted cells was isolated using the PicoPure RNA Isolation Kit (Life Technologies, KIT0214). Reverse transcription was performed with the High-Capacity cDNA Reverse Transcription Kit (Life Technologies, 4368814) or the SuperScript IV VILO Master Mix (Life Technologies, 11766050). qRT-PCR was executed using Power SYBR Green PCR Master Mix (Life Technologies, 4368708) and the primers listed in Table 1. Relative expression levels were calculated with the $2^{-\Delta\Delta Ct}$ method after normalization to the house keeping genes *Cph* or *Hprt*.

### Western blot

For Western blot analysis, cells were lysed with the Cell Lysis Buffer (Cell Signaling, 9803S). Subsequentially, samples were run on a SDS-PAGE and blotted onto PVDF membranes (Merck Chemicals, ISEQ00010) overnight. After blocking, primary antibodies were

incubated overnight in 5% milk-TBST using the following concentrations: anti-HAS2 (1:500, Abcam, ab140671), anti-VCP (1:5000, Abcam, ab109240), anti-ACTB (1:5000, Sigma Aldrich, A5441), anti-LRRC15 (1:1000, Abcam, ab150376), anti-E-Cadherin (1:2000, BD Biosciences, 610181), anti-HAPLN1 (1:1000, Bio-Techne, AF2608), anti-pP65 (Ser536) (Cell Signaling, #3033), anti-pFAK (Tyr925) (Cell Signaling, #3284), anti-pSTAT3 (Tyr705) (Cell Signaling, #9145). Secondary antibodies goat anti-rabbit HRP, rabbit anti-mouse HRP (1:2500, DAKO, P0448, P0260) or donkey anti-goat HRP (1:2500, R&D, HAF109) were incubated for 1 h prior to detection with SuperSignal West Pico Plus Chemisubstrate (VWR International, PIER34577).

### Cell culture maintenance

All cells used for in vitro culture were cultured in DMEM, low glucose (high glucose for HEK293A + T cells), GlutaMAX Supplement (Life Technologies, 21885108) supplemented with 10% FCS (VWR International, S181B) and 1% Pen/Strep (Life Technologies, 15140122). For 2D experiments, cells were starved for 48 h prior to analysis. Cells were periodically analyzed by PCR for mycoplasma contamination (primers: gggagcaaacaggattagatataccct, tcggaccatcatctgtcactctgttaac ctc). Authentication was not deemed necessary.

### Mouse experiments

For mouse experiments, female C57BL/6J mice were ordered from Janvier at 9–11 weeks of age. 1 million KPC or KPC-HAPLN1 cells expressing luciferase and RFP were injected intraperitoneally (i.p.) into mice. At day 8 and 10 of tumor growth, the belly of mice was shaved and Beetle luciferin (Promega, E1603) was injected i.p. 20 min after injection, luminescence was measured using an IVIS® Lumina LT In Vivo Imaging System. In total, tumors grew for 11 days. Mice were euthanized by rapid cervical dislocation. Peritoneal lavage was performed to remove cells suspended in the peritoneum by injection and subsequent removal of PBS into the peritoneal cavity after euthanization. Solid tumor masses formed in tumor cell injected mice, their size was always under the regulation limit (10 mm of diameter). For flow cytometry and cell sorting, solid tumors were digested using 2,5 mg/ml collagenase D (Sigma Aldrich, 11088866001), 0,5 mg/ml Liberase DL (Sigma Aldrich, 5466202001) and 0,2 mg/ml DNase (Sigma Aldrich, 10104159001) in 10% FCS DMEM for 45 min at 37 °C after mincing thoroughly. If peritoneal lavage contained erythrocytes, red blood cell lysis was performed with RBC Lysis Buffer (Life Technologies, 00-4333-57).

### Flow cytometry and cell sorting

Analysis by flow cytometry was performed on a BD LSR Fortessa, while BD FACS Aria/BD FACS Aria Fusion Sorters were used for cell sorting. The listed antibodies or dyes in Table 2 were incubated for 30 min with the cells before washing and analysis/sorting. For intracellular staining, cells were first stained with extracellular staining, then fixed, permeabilized and intracellular staining for 20 min was performed. Cell cycle was analyzed in fixed cell (70% ethanol) by permeabilization and DNA staining with propidium iodide (5 μg/mL, Sigma-Aldrich, 537059) in presence of RNase A (1 mg/mL, Sigma-Aldrich, R5125). Compensation was carried out with UltraComp eBeads Compensation Beads (Life

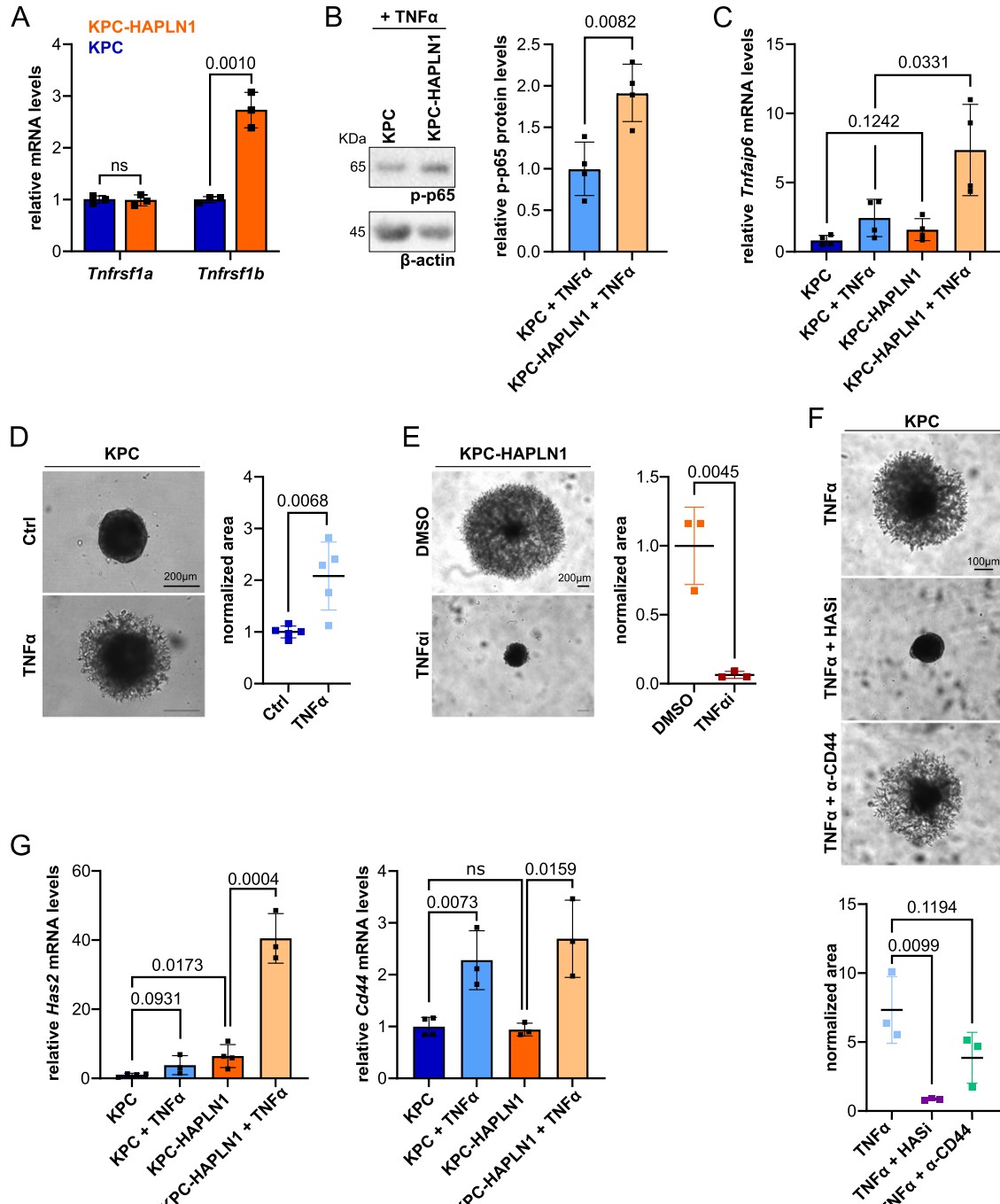

**Fig. 5 | Increased TNF signaling induces tumor cell plasticity. A** KPC (blue) and KPC-HAPLN1 (orange) cells were cultured and TNFα receptors expression analyzed by qPCR. $n = 3$. **B** Western blot analysis of phosphorylated p65 (p-p65) after 15 min of 50 ng/ml TNFα stimulation as readout for increased NFκB signaling. β-actin was used as loading control. $n = 4$. **C** Analysis of TNF-induced gene expression in vitro. Stimulation with 50 ng/ml recombinant TNFα for 6 h in starvation. $n = 4$. **D** Invasion assay into Matrigel of KPC cells stimulated with 50 ng/ml TNFα. Invasion was measured as relative occupied area 48 h after embedding. $n = 5$. Scale bar: 200 μm.

**E** Invasion assay of KPC-HAPLN1 cells into Matrigel using an TNFα antagonist (10 μg/ml). $n = 3$. **F** Invasion assay of KPC cells into Matrigel after addition of 50 ng/ml TNFα, 500 μM 4-methylumbelliferrone (HASi) or anti-CD44 blocking antibody. Pictures taken 48 h after embedding and stimulation. $n = 3$. Scale bar: 100 μm. **G** Relative mRNA levels of *Has2* and *Cd44* after 6 h of 50 ng/ml TNFα stimulation. KPC $n = 4$; rest $n = 3$. Graphs represent mean ± SD. Data points indicate independent biological replicates. Unpaired two-tailed T test was used for statistical analysis.

Technologies, 1-2222-42). FlowJo Software was used for analysis of the acquired samples.

### Generation of (stable) cell lines

Stable overexpression was achieved by lentiviral infection of KPC and PANC-1 cells (ATCC). For HAPLN1 overexpression the sequence coding for HAPLN1 was inserted into a pLenti6-V5dest plasmid. The plasmid

pLL3.7 (Addgene #11795) was used for eGFP, while EF.CMV.RFP plasmid (Addgene #17619) or pLenti-RFP-Puro (Hölzel, LTV-403) was used for RFP labeling. The plasmid of interest was transfected together with the plasmids for packaging (psPAX2) and envelope (pMD2, encoding for VSV-G) into HEK293T cells. Cells were infected and successful integration into the genome was guaranteed by selection with Blasticidin-Hydrochloride (Sigma Aldrich, 15205) for a minimum of

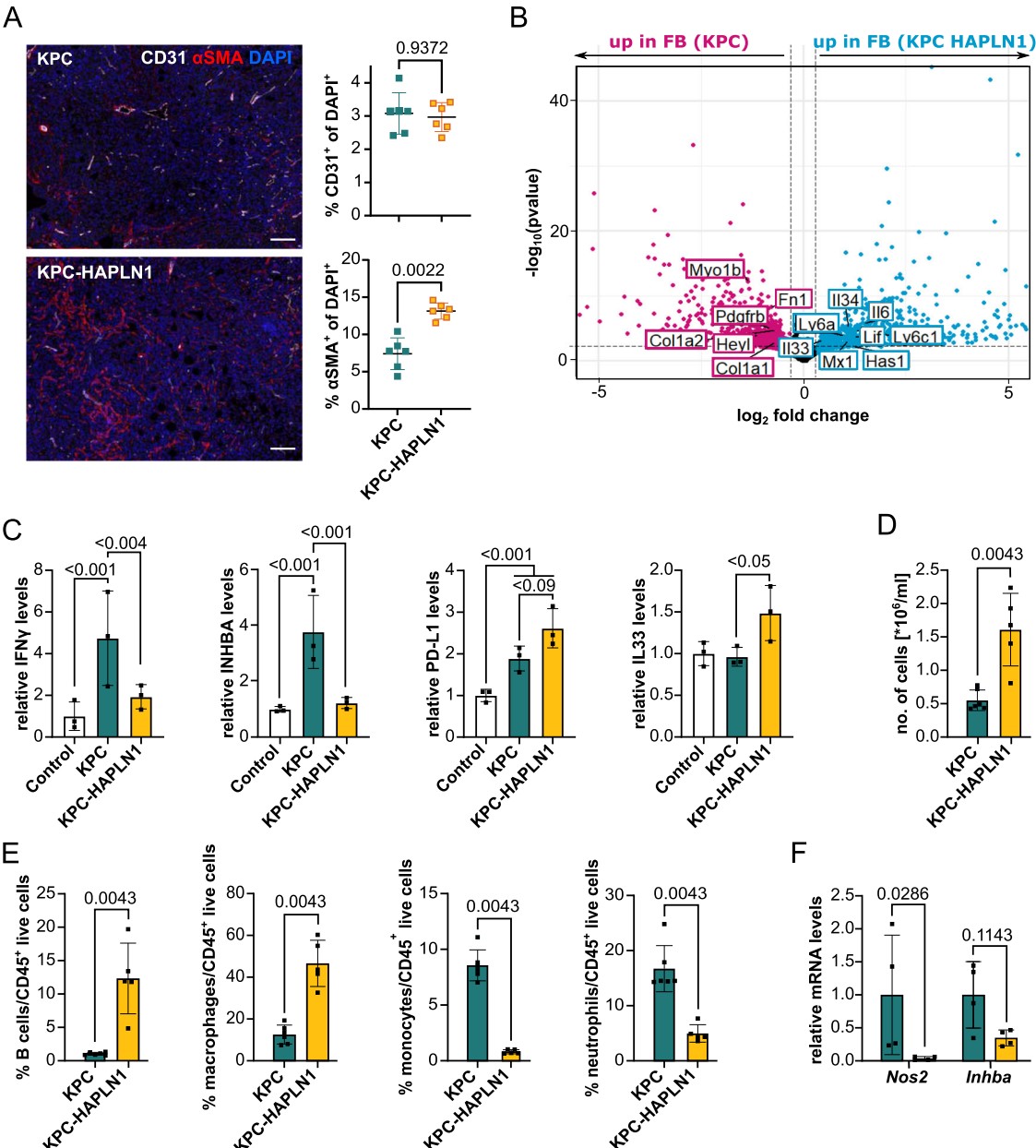

**Fig. 6 | HAPLN1 modifies the immune microenvironment. A** CD31/αSMA immunofluorescence staining on tumors. Representative images and quantification of CD31⁺ cells and αSMA⁺ cells are shown. Scale bar: 50 μm. *n* = 6. **B** RNA sequencing of PDPN⁺ isolated cancer-associated fibroblasts from KPC or KPC-HAPLN1 tumors. Pink dots mark genes significantly up in CAFs from KPC, blue dots genes significantly up in CAFs from KPC-HAPLN1 tumors. Some genes of interest are labeled. *n* = 3. **C** Serum of healthy, or KPC/KPC-HAPLN1-injected mice at day 11 of tumor growth was analyzed using the scioCD protein array. Relative protein levels displayed. *n* = 3. **D** Number of cells per milliliter in the peritoneal lavage of mice after 11 days of tumor growth. KPC *n* = 6; KPC-HAPLN1 *n* = 5. **E** Immune cell populations in the peritoneal lavage assessed by flow cytometry. KPC *n* = 6; KPC-HAPLN1 *n* = 5. **F** Macrophages were isolated from the peritoneum by cell sorting. qRT-PCR analysis of inflammatory marker genes. *n* = 4. Graphs represent mean ± SD. Data points indicate independent biological replicates. Non-parametric two-tailed Mann–Whitney U test was used.

10 days. For control cells, an empty pLenti6-V5dest plasmid was used. eGFP and RFP positive cells were sorted out to receive a homogeneously marked population.

For adenoviral infection, plasmids encoding for eGFP and mCherry were amplified in HEK293A cells and cells of interest were infected afterwards. Analysis was carried out no later than 96 h post infection to prevent loss of overexpression.

**Spheroid formation and invasion assay**
For 3D culture, 5.000 cells were seeded in a 96 well ultra-low attachment plate (NEO Lab, 174929) in DMEM + 10% FCS. After 96 h, pictures of the formed spheroids were taken with a Zeiss Cell Observer. For analysis by mRNA or flow cytometer, cells were seeded in 6 well plates coated with Poly(2-hydroxyethyl methacrylate) (Poly-HEMA, Sigma Aldrich, P3932) and afterwards lysed, or digested with 0,05% Trypsin (Life Technologies, 25300054) for 5 min respectively. Subsequential to digestion, cells were stained with DAPI for analysis by flow cytometry. For invasion assay, spheroids were formed for 48 h in 96 well plates, before half of the medium was replaced with Matrigel (Fisher Scientific, 11523550) or Collagen extracted from rat tails. When stimulating the spheroids, they were treated with HA (mix of LMW and HMW HA, both at 5 μg/ml; R&D Systems, GLR001 and GLR002), recombinant

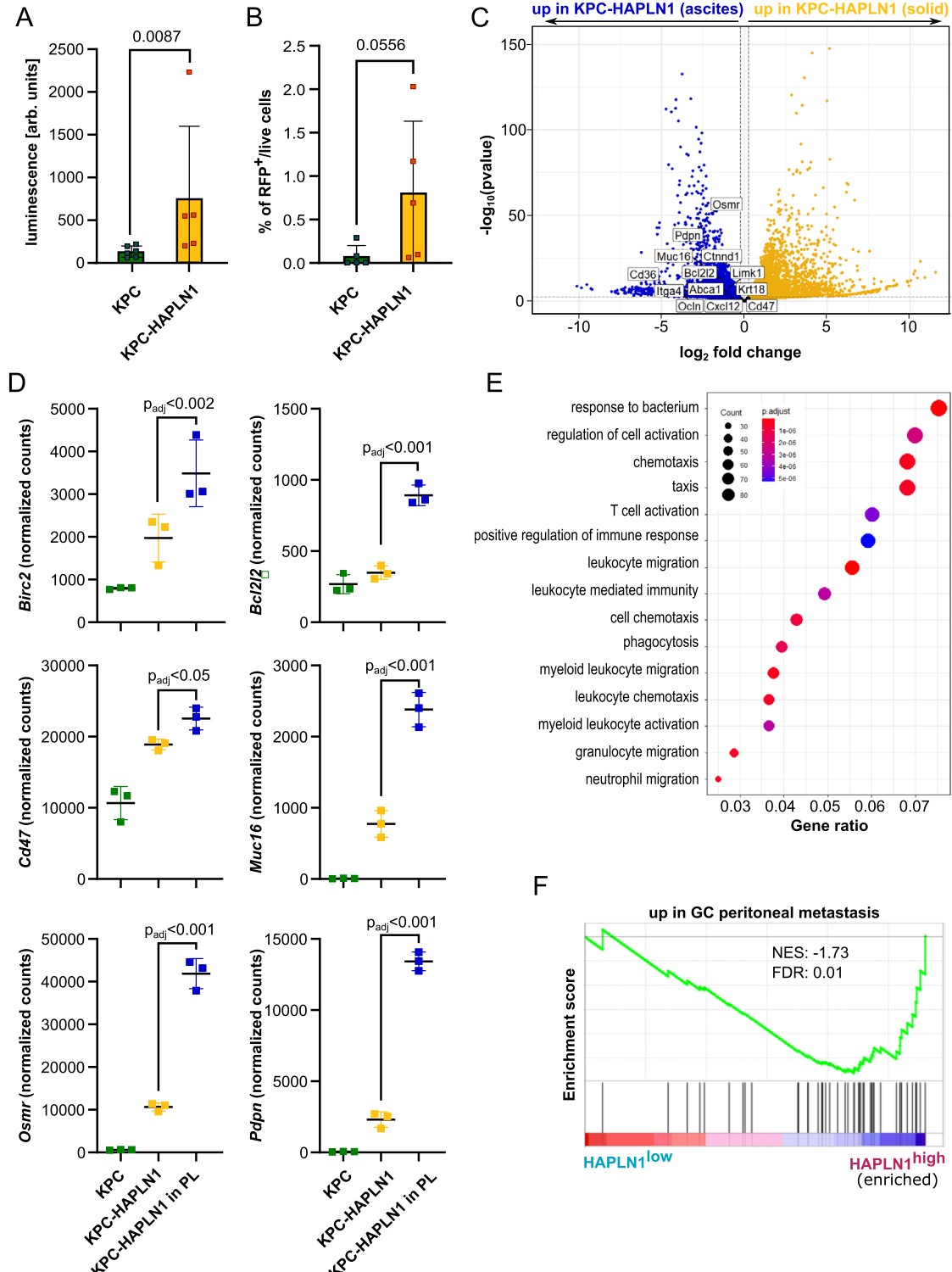

**Fig. 7 | HAPLN1 facilitates peritoneal colonization by tumor cells. A** Ex vivo luminescence measurement on peritoneal lavage of KPC and KPC-HAPLN1 mice 11 days after tumor injection. KPC $n = 6$; KPC-HAPLN1 $n = 5$. **B** RFP+ tumor cells detected by flow cytometry in the peritoneal lavage of mice at day 11. $n = 5$. **C** RNA sequencing results comparing KPC-HAPLN1 cells from solid tumor or in suspension in peritoneal lavage. Blue dots mark genes upregulated in cells in suspension, yellow dots genes upregulated in cells in tumor mass. Some genes of interest are highlighted $n = 3$. **D** Selected genes displayed using normalized read counts. $n = 3$. **E** Gene Ontology of 'Biological Process' on genes upregulated in KPC-HAPLN1 cells

which detached into the peritoneal cavity. **F** GSEA on the PDAC patient data set of Cao et al. (2021), divided in high or low *HAPLN1* expression by the mean *HAPLN1* expression. A gene set for genes upregulated in gastric cancer (GC) peritoneal metastasis was used. $n = 140$. Graphs represent mean ± SD. Data points indicate independent biological replicates. Non-parametric two-tailed Mann−Whitney $U$ test was used for panel **A** and **B**. For panels **C**–**E**, the package DESeq2 in R studio was used. In this package, the Wald test is used for hypothesis testing when comparing two groups. Correction was performed by multiple testing using the Benjamini−Hochberg (BH) method.

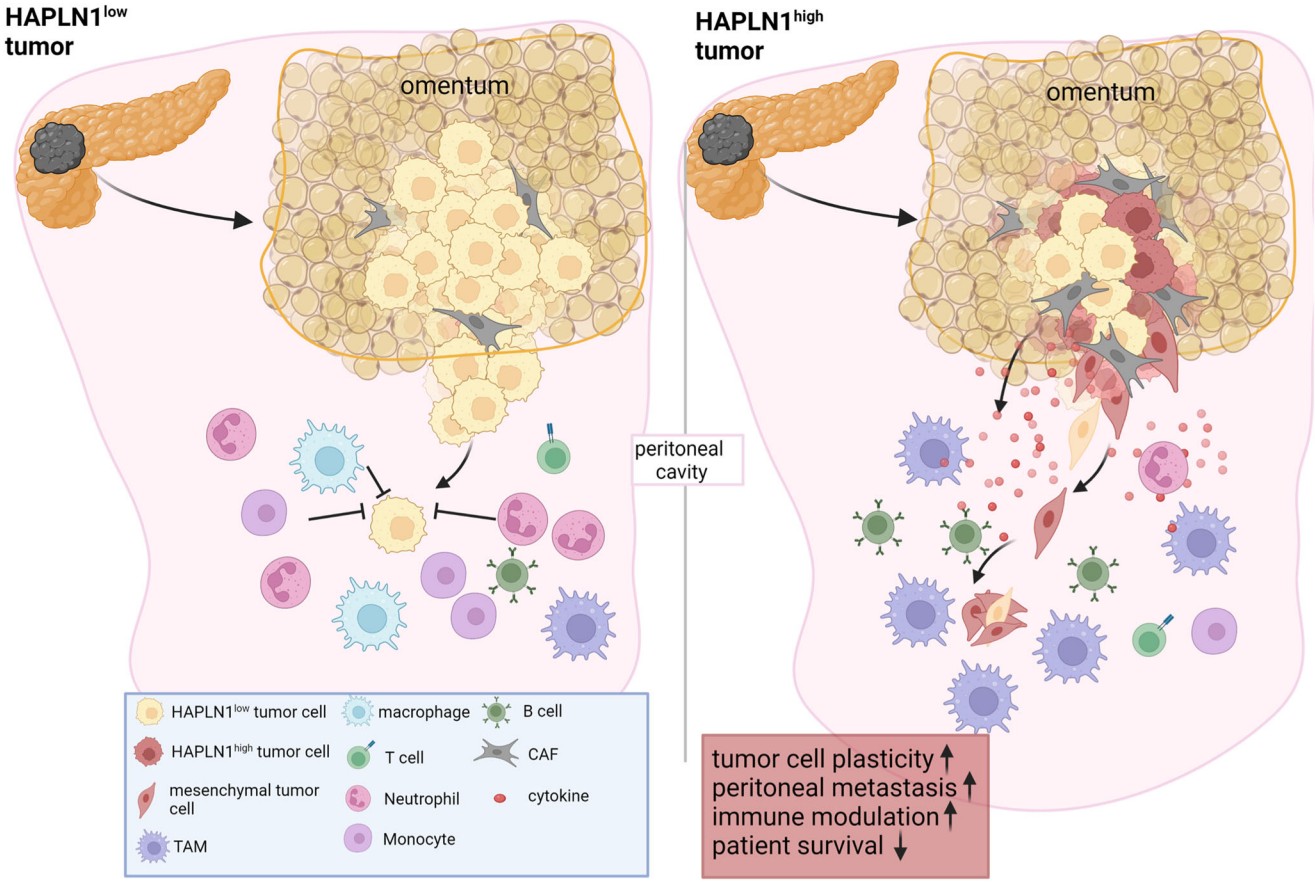

**Fig. 8 | HAPLN1 drives peritoneal carcinomatosis by inducing cell plasticity.** We propose that in HAPLN1 low conditions tumor cells are less equipped to survive in the peritoneal cavity due to a decreased migratory capacity and a tumoricidal environment. When HAPLN1 expression increases, cells become more plastic, increasing invasiveness and preconditioning the peritoneal cavity for an increased tolerance of invading tumor cells.

HAPLN1 (80 ng/ml, BioTechne, 2608-HP-025), Hyaluronidase (200 mg/ml, MP Biomedicals, 151272), recombinant TNFα (50 ng/ml; PeproTech, murine: 315-01A-20, human: 300-01A-50), IgG control antibody (1 µg/ml. Biozol Diagnostica, BXC-BE0090-1MG), anti-CD44 blocking antibody (1 µg/ml. Biozol Diagnostica, BXC-BE0039-1MG), or TNFα inhibitor (10 µg/ml, Merck, 654256), immediately before the addition of matrigel or collagen. When using the following inhibitors spheroids were pre-treated for 24 h with (HASi (4-Methylumbelli-ferrone (4-MU), 500 µM, Sigma Aldrich, M1381), FAKi (GSK2256098, 10 µM, Cayman, Cay22955-1), STAT3i (Stattic, 25 µM, Abcam, ab120952), PI3Ki (LY294002, 10 µM, BioTrend, HY-10108), MEKi (PD98059, 50 µM, New England Biolabs, 9900 L), ERKi (ERK Activation Inhibitor Peptide I, 1 µM, Sigma Aldrich, 328000) or DMSO as control, before embedding. For cocultures, equal amounts of KPC and KPC-HAPLN1 cells expressing either RFP/mCherry or GFP were seeded. 48–120 h after addition of Matrigel or collagen, pictures of invasion were taken by a Zeiss Cell Observer or a Zeiss LSM710 ConfoCor3.

## BMDM extraction
Bone marrow progenitors were isolated from the leg bones (tibia and femur) of wildtype mice by flushing the bones with PBS. Cells were then cultured on untreated plates for 7 days adding 10 ng/ml recombinant murine M-CSF (Peprotech, AF-315-02-100) at day of isolation. Afterwards, cells were counted and seeded for experiments.

## Conditioned matrix experiment
For conditioned matrix experiments, KPC or KPC HAPLN1 cells were seeded in 6 well plates and cultured for 3 days. Then, by the addition of puromycin, cells were killed over several days until no living cell was

left. Medium was removed and plates washed 3x 10 min with sterile PBS, before GRX fibroblasts were plated on top for 24 h before lysis for mRNA analysis.

## Immunofluorescence and immunohistochemistry
For staining of formalin-fixed paraffin-embedded mouse and human tumors, sections were deparaffinated. Antigen retrieval in pH = 9 was used for HAPLN1, pSTAT3 and CD31 stainings, pH = 6 for CD45 staining and treatment with Proteinase K for PDPN staining. Afterwards, slides were blocked, and then incubated ON with anti-CD31 (1:50, Abcam, ab28364), anti-pSTAT3 (1:200, Cell Signaling, #9145), anti-CD45 (1:50, BD Bioscience, 553076), anti-HAPLN1 (1:200, Bio-Techne, AF2608) or anti-PDPN (1:400, Abcam, ab11936). After washing, for immuno-fluorescence stainings, slides were incubated for 2 h at RT with anti-αSMA antibody (1:200, Sigma Aldrich, C6198-100UL) and Goat anti Rabbit Alexa Fluor 647 (1:200, Life Technologies, A21245), Goat anti Rat Alexa Fluor 647 (1:200, Life Technologies, A21247), or Goat Anti-Syrian hamster IgG H&L Alexa 647 (1:200, Abcam, ab180117). After washing, nuclei were stained using DAPI at 2,1 nM. For DAB staining of HAPLN1, slides were treated for 10 min with 3% $H_2O_2$ after incubation of the primary antibody. After washing, 30 min incubation with an anti-goat HRP antibody (BioTechne, VC004-025) was performed before incubation with DAB (Biozol Dignostica, ZYT-DAB057) and counter-staining with hematoxylin. Pictures were taken with the Zeiss AxioS-can.Z1. Quantification was performed using Fiji software.

## Scratch assay
For the scratch assay, 100.000 cells were seeded into the wells of a cell view cell culture slide (Greiner Bio-One, 543078). The next day,

## Table 1 | Primers for qRT-PCR

| Gene name | Fw primer 5'–3' | Rev primer 5'–3' |
|---|---|---|
| Abca1 | CGTGTCTTGTCTGAAAAAGGAGG | CGTGTCACTTTCATGGTCGC |
| Abcg1 | CTTTCCTACTCGTGTACCCGAGG | CGGGGCATTCCATTGATAAGG |
| Acan | CCTGCTACTTCATCGACCCC | AGATGCTGTTGACTCGAACCT |
| Arg1 | CAGAAGAATGGAAGAGTCAG | CAGATATGCAGGGAGTCACC |
| Cd274 | GCTCCAAAGGACTTGTACGTG | TGATCTGAAGGGCAGCATTTC |
| Cdh1 | CAGGTCTCCTCATGGCTTTGC | CTTCCGAAAAGAAGGCTGTCC |
| Col1a2 | TGGTGATAAAGGGCACAGGG | ACCATGTAGGCCAGCAAGAC |
| Cph | ATGGTCAACCCCACCGTG | TTCTTGCTGTCTTTGGAACTTTGTC |
| Cxcl12 | TGCATCAGTGACGGTAAACCA | TTCTTCAGCCGTGCAACAATC |
| H2-Ab1 | AGCCCCATCACTGTGGAGT | GATGCCGCTCAACATCTTGC |
| Has1 | GGCGAGCACTCACGATCATC | AGGAGTCCATAGCGATCTGAAG |
| Has2 | TGTGAGAGGTTTCTATGTGTCCT | ACCGTACAGTCCAAATGAGAAGT |
| Has3 | CCTGGAGCACCGTCGAATG | CCTTGAGGTTTGGAAAGGCAA |
| Hprt | TGTTGTTGGATATGCCCTTG | ACTGGCAACATCAACAGGACT |
| HPRT | CCTGGCGTCGTGATTAGTGAT | AGACGTTCAGTCCTGTCCATAA |
| L1cam | AAAGGTGCAAGGGTGACATTC | TCCCCACGTTCCTGTAGGT |
| Lif | ATTGTGCCCTTACTGCTGCTG | GCCAGTTGATTCTTGATCTGGT |
| Lrrc15 | CAGGTTTGGCCTACTATGGCT | GGGGAGTTCGGTGATGTGTG |
| Plod2 | GAGAGGCGGTGATGGAATGAA | ACTCGGTAAACAAGATGACCAGA |
| Snai2 | TGGTCAAGAAACATTTCAACGCC | GGTGAGGATCTCTGGTTTTGGTA |
| Tnf | CCACGTCGTAGCAAACCACC | GATAGCAAATCGGCTGACGG |
| Tnfaip3 | GAACAGCGATCAGGCCAGG | GGACAGTTGGGTGTCTCACATT |
| Tnfaip6 | GGGATTCAAGAACGGGATCTTT | TCAAATTCACATACGGCCTTGG |
| TNFAIP6 | TTTCTCTTGCTATGGGAAGACAC | GAGCTTGTATTTGCCAGACCG |
| Tnfrsf1a | CCGGGAGAAGAGGGATAGCTT | TCGGACAGTCACTCACCAAGT |
| Tnfrsf1b | ACACCCTACAAACCGGAACC | AGCCTTCCTGTCATAGTATTCCT |
| Twist1 | GGACAAGCTGAGCAAGATTCA | CGGAGAAGGCGTAGCTGAG |
| Vcan | ACTAACCCATGCACTACATCAAG | ACTTTTCCAGACAGAGAGCCTT |

## Table 2 | Flow cytometry and cell sorting antibodies

| Target with conjugate | Dilution | Company | Cat. No. |
|---|---|---|---|
| **B220**-APC | 1:100 | BioLegend | 103211 |
| **CD11b**-BUV805 | 1:800 | BD Biosciences | 741934 |
| **CD11b**-PE-Cy7 | 1:800 | BD Biosciences | 552850 |
| **CD19**-Alexa700 | 1:50 | BD Biosciences | 557958 |
| **CD25**-PE | 1:200 | Life Technologies | 12-0251-82 |
| **CD31**-APC | 1:100 | BD Biosciences | 561814 |
| **CD3**-PerCp-Cy5.5 | 1:50 | BioLegend | 100217 |
| **CD45**-PerCp-Cy5.5 | 1:200 | BD Biosciences | 550994 |
| **CD45**-FITC | 1:400 | BD Biosciences | 553079 |
| **CD4**-PE-Cy7 | 1:300 | Life Technologies | 25-0041-82 |
| **CD8**-APC-Cy7 | 1:100 | BD Biosciences | 557654 |
| **DAPI** (stock: 10,5 mM) | 1:10.000 | Life Technologies | D3571 |
| **F4/80**-Alexa Fluor 488 | 1:100 | Life Technologies | 53-4801-82 |
| **FoxP3**-Alexa Fluor 488 | 1:50 | BD Biosciences | 560403 |
| **Ly6C**-PE | 1:200 | BD Biosciences | 560592 |
| **Ly6G**-APC-Cy7 | 1:100 | BD Biosciences | 560600 |
| **MHC-II**-BV510 | 1:100 | BD Biosciences | 742893 |
| **PDPN**-APC-Cy7 | 1:100 | BioLegend | 127417 |
| **SiglecF**-BUV395 | 1:100 | BD Biosciences | 740280 |

cells were starved and a small pipette tip was used to generate a scratch through the well. Immediately after, cells were placed into a Zeiss Cell Observer with incubation chamber, keeping the conditions at 5% $CO_2$ and 37 °C. Pictures were taken every 30 min for a total time span of 15 h. Migration of the cells was analyzed with the Fiji Software.

### ScioCyto cytokine array of serum
Blood was taken from the *vena facialis* 11 days after tumor injection and collected in a microvette LH (Sarstedt, 201.345) to receive the serum. Sciomics GmbH running the scioCyto Cytokine array performed the analysis of the serum of tumor bearing and control mice.

### Ex vivo luminescence
For ex vivo luminescence was measured on cells of the peritoneal lavage using Luciferase Assay System (Promega, E1500) and a CLARIOStar plate reader.

### RNA sequencing
Libraries and multiplexes were prepared with NEBNext Multiplex Oligos for Illumina (New England Biolabs, E7600S) and NEBNext Single Cell/Low Input RNA Library Prep Kit for Illumina (New England Biolabs, E6420S). NextSeq 550 PE 75 HO (Illumina) RNA sequencing was performed by the Genomics & Proteomics Core Facility of the German Cancer Research Center (DKFZ, Germany).

### Classification in HAPLN1[high] and HAPLN1[low] cohorts for survival, GSEA and cellular content
For the survival data we made use of the publicly available transcriptomic, proteomic and clinical data published in Cao et al. 2021[23]. For the distribution of patients into the HAPLN1[high] or HAPLN1[low] cohorts, we extracted the *HAPLN1* expression (obtained by RNA-sequencing) or HAPLN1 protein levels (obtained by mass

spectrometry) and calculated the mean expression levels over all patients. Patients were assigned to the HAPLN1[high] cohort, if their HAPLN1 expression level was above this mean, and vice versa. mRNA and protein analyses were conducted independently. Then we ordered patients according to follow up days and vital status, and plotted resulting survival curves. Same classification was employed for the analysis of the cellular content of the tumors, using the clinical data provided by the authors of the dataset.

## Gene Set Enrichment Analysis (GSEA)

To investigate differentially expressed genes, Gene Set Enrichment Analysis (GSEA,[54]) was performed. p values were assessed using 1000 permutations for each gene set and corrected with the false discovery rate (FDR) method. When analyzing microarray data, the mean of all probes for the same gene was used to divide patients into high/low cohorts. The following publicly available data sets of PDAC patients were used for our analyses: NCBI-GEO GSE62452 and the data from Cao et al.[23,24].

## Bioinformatic analysis of RNA-sequencing data

Bioinformatics analysis of RNA-sequencing data sets was performed using the software R using the DESeq2 package. Plots were created using ggplot2 or EnhancedVolcano. apeglm package was used for LFC shrinkage.

## Statistics

Graphs represent mean ± SD with each data point depicting a biological replicate, the n in the legends represents the number of biologically independent experiments. Statistical analysis was performed using the GraphPad Prism 9 software. For in vitro experiments, Gaussian distribution was assumed, and two-tailed parametric unpaired $t$-test was applied, while the non-parametric Mann–Whitney test was used for in vivo and ex vivo experiments. Data was considered as statistically significant if $p < 0.05$. For high throughput analysis (RNA-seq, scioCyto Cytokine array) $p_{adj}$ was calculated and results were considered significant if $p_{adj} < 0.05$.

## Schemes

Schemes in Figs. 2I, 8 and Supplementary Fig. 3A were created with BioRender.com.

## Reporting summary

Further information on research design is available in the Nature Portfolio Reporting Summary linked to this article.

## Data availability

All data supporting the findings of this study are available within the paper and its Supplementary Information. Source data are provided as a Source Data file RNAseq data are available from the Gene Expression Omnibus under accession number GSE211126. Source data are provided with this paper.

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

## Acknowledgements

We would like to thank Stephen Konieczny, from Purdue University, Indiana, for providing us the KPC cell line. Additionally, we thank the Light Microscopy core facility, the High Throughput Sequencing Unit of the Genomics and Proteomics Core Facility and the Omics IT and Data Management (ODCF), the Flow Cytometry core facility and animal caretakers of the German Cancer Research Center (DKFZ) for providing excellent services. We would like to thank Damir Krunic (DKFZ, Light Microscopy Core Facility) for his help with FIJI software data analysis.

This work was funded by the Deutsche Forschungsgemeinschaft (DFG) project KFO5002 and the German Ministry of Education and Research within the SATURN3 consortium project 01KD2206N (to L.-C.C.). DFG project 394046768 - SFB1366 projects C4 and Z2 (to A.F. and CM). DFG project 419966437, Deutsche Krebshilfe project 70113888, Early Career Research Award by the MasQueUnTrail Association and the Spanish Association for Cancer Research, MCIN/AEI/ 10.13039/ 501100011033 (PID2020-117946GB-I00 and RYC2019-027937-I) (to J.R-V.). The Science Ministry of Spain or the Health Ministry (ISCIII) receives support from the EU and its ERDF program. Part of the equipment used in this work has been funded by Generalitat Valenciana and co-financed with ERDF funds (OP ERDF of Comunitat Valenciana 2014-2020).

## Author contributions

Conception and design: L.W., F.D-R., A.F., J.R-V.; Acquisition of data: L.W., F.D-R., N.V-S., E.D., I.M., C.M., H.B., E.F-F., M.V., J.G.; Analysis and interpretation of data: L.W., F.D-R., N.V-S., E.D., E.E., E.A-S., C.M., H.B., L.-C.C., E.F-F., R.M., A.T., A.F., J.R-V.; Initial draft: L.W.; Revision of the manuscript: L.W., F.D-R., A.F., J.R-V.,; Study supervision: A.F., J.R-V.

## Competing interests

The authors declare no competing interests.

## Additional information

[1]Division Vascular Signaling and Cancer, German Cancer Research Center (DKFZ), 69120 Heidelberg, Germany. [2]Faculty of Biosciences, University of Heidelberg, 69120 Heidelberg, Germany. [3]Tumor-Stroma Communication Laboratory, Centro de Investigación Príncipe Felipe, 46012 Valencia, Spain. [4]Division of Stem Cells and Cancer, German Cancer Research Center (DKFZ), 69120 Heidelberg, Germany. [5]HI-STEM - Heidelberg Institute for Stem Cell Technology and Experimental Medicine gGmbH, 69120 Heidelberg, Germany. [6]Department of Pathology and Experimental Therapy, School of Medicine, University of Barcelona (UB), L'Hospitalet de Llobregat, Barcelona, Spain. [7]Molecular Mechanisms and Experimental Therapy in Oncology Program (Oncobell), Institut d'Investigació Biomèdica de Bellvitge (IDIBELL), L'Hospitalet de Llobregat, Barcelona 08908, Spain. [8]Institute for Clinical Chemistry, University Medical Center Göttingen, 37075 Göttingen, Germany. [9]Institute of Pathology, University Medical Center Göttingen, Georg-August-University, 37075 Göttingen, Germany. [10]Clinic of General, Visceral and Pediatric Surgery, University Medical Center Göttingen, Robert-Koch-Straße 40, 37075 Göttingen, Germany. [11]Institute of Pathology, Technical University of Munich, 81675 Munich, Germany. [12]German Center for Cardiovascular Research (DZHK), partner site Göttingen, Germany. [13]These authors contributed equally: Lena Wiedmann, Francesca De Angelis Rigotti. [14]These authors jointly supervised this work: Andreas Fischer, Juan Rodriguez-Vita. ✉e-mail: andreas.fischer@med.uni-goettingen.de; jrodriguez@cipf.es

