## [Peer Review File · Nature Communications]

HAPLN1 potentiates peritoneal metastasis in pancreatic cancerEditorial Note: Parts of this Peer Review File have been redacted as indicated to remove third-party material where no permission to publish could be obtained.

REVIEWER COMMENTS

Reviewer #1 (Remarks to the Author):

Wiedmann et al. demonstrate HAPLN1 is enriched in the basal/mesenchymal PDAC and that functionally, expression of HAPLN1 leads to immunomodulation and increased peritoneal spread in a mouse model. Overall, peritoneal spread is a common path for PDAC metastasis that has significant effects on survival and a major source of morbidity. This study evaluates this significant problem and is important for the field. The work is done well with supportive human correlative data and preclinical models.

Comments/Questions

1) Introduction

a. The authors use EMP and EMT in the intro, but I would stick with EMT since this is what the field uses. EMP is certainly both EMT and MET so you could also just indicate EMT and MET have been shown and this represents plasticity.

b. The section on omental metastasis followed by peritoneal metastasis makes it seem that this is the known path to the peritoneum, but I don't think this is known. I would rephrase to say that omental and peritoneal metastatic spread appears to be driven by a combination of niche and EMT

2) Figure 1: Analysis of published expression profiles indicates HAPLN1 is associated with EMT and survival. This analysis is done well and I have no additional comments. One could also do a similar analysis on other data sets (TCGA, ICGC, etc.) to gain further confidence

3) Figure 2: HAPLN1 overexpression in KPC cell lines shows higher HA, Has2, Vcan and EMT markers. Overall, experiments are reasonable. ? if there is any KPC line or growth conditions they have found with high endogenous HAPLN1 expression. This would probably be helpful to support physiological relevance of HAPLN1.

4) Figure 3: I'm surprised by the inverse effect of MEKi on invasiveness. Most of the literature has indicated that KRAS->MEK signaling is a driver of EMT. Is this something unique to this cell line or this particular MEK inhibitor? Additional cell lines or other MEK inhibitors or CRISPR for MAPK would help determine this discrepancy with the broader literature.

5) Figure 4: In vivo data showing changes in PDAC and CAF cells. RNA-seq in PDAC cells shows loss of Acan1 c/w in vitro and gain of EMT genes. Overall the data is done well but consider putting KPC vs KPC-HAPLN1 directly on the volcano plots to improve clarity.

6) Figure 5/6: Immune and metastasis in animal data. Overall really well done. ? if the authors can expand in the discussion thoughts on the relative contribution of HAPLN1 metastasis to immune/Macrophage changes vs EMT changes?

Reviewer #2 (Remarks to the Author):

In this work the authors explored the contribution of HAPLN to peritoneal metastasis by pancreatic ductal adenocarcinoma (PDAC). Because HAPLN is known to be important in other tumor contexts, expression was evaluated in publicly available datasets which showed increased HAPLN in PDAC primary tumors (vs normal) and shortened overall survival times. To investigate HAPLN experimentally, the authors (over)-expressed HAPLN in a KPC cell line (HAPLN^{high}). Over-expressing HAPLN resulted in gene expression and spheroid changes suggestive of increased HA production, cell plasticity, and invasion. i.p. injection of tumor cells showed that CAFs and the immune landscape in tumors formed by HAPLN^{high} diverged from HAPLN^{low}, and that HAPLN^{high} cells more efficiently colonized the peritoneal cavity than HAPLN^{low} cells.

Although intriguing and potentially of high importance, the data are descriptive, lack appropriate functional interrogation, and no mechanistic insights are provided for how HAPLN might cause the observed changes or promote peritoneal-specific metastasis (as opposed to distant/hematogenous metastasis or primary tumor growth). In my opinion the work as it stands is preliminary would require additional experimentation to achieve suitable rigor and mechanistic insights, as outlined below.

General Points:

1. Figure 1: The authors report that HAPLN^{high} is enriched in public data of patient primary tumors, including subtypes with a basal-like signature. However, it is unclear if expression is from tumor cells themselves, from stroma constituents (e.g. CAFs), or both. Are these samples bulk tumor tissues or microdissected? This is especially pertinent considering primary PDACs are often very stroma rich, and other studies have shown HAPLN is secreted into stroma by CAFs.

2. Figures 2-6: All experimental data is based on a KPC cell line with forced over-expression of HAPLN. This is a very limited reagent set (one cell line) using a very limited approach (over-expression). KPC mice themselves are well-known to spontaneously develop peritoneal metastasis and malignant ascites fluid (from primary tumors). The work would be strengthened by examination of HAPLN in the natural context of peritoneal metastasis complemented by loss-of-function approaches from cells isolated from these or similar such lesions.

3. Figures 2-6: It seems the authors assume that HAPLN must mediate all the observed effects through linking cells to HA and/or stimulating HA synthesis/production. However, this is never addressed mechanistically or tested functionally. It is unclear how one protein or a single protein:matrix interaction would mediate all the observed effects. This is especially relevant considering that HAPLN can be either a tumor suppressor or oncogene depending on the context, and PEGPH20 has not fared well in clinical trials for PDAC patients.

Technical Points

1. Figure 1F: The survival curves for mRNA and protein seem strikingly similar. Please provide more clarity on the methodology behind these plots, especially how the cut-off between HAPLN^{high} and HAPLN^{low} based on mean values was determined. Was the protein data extracted from pathology scoring (by IHC), digital quantification, mass spectrometry, or some other method? I could not find any of this information in the legend or the methods.

2. Figure 2C: The authors show that KPC cells are HAPLN^{low}, necessitating forced over-expression of HAPLN. However, as pointed out above KPC mice develop peritoneal metastasis and the authors also show high HAPLN expression in patient primary tumors. Can the authors address these discrepancies?

3. Figure 3: Rescue of inhibitor effects (with exogenous HA for example) and phenocopy of inhibitor effects (with CD44 knockdown, HAPLN knockdown, hyaluronidase Tx's, as examples) would greatly strengthen the confidence that HA/CD44/HAPLN mediate the observed spiky ("invasion") morphology observed in matrigel.

4. Figure 4: The authors detect enrichment for inflammatory-type CAFs ("iCAFs") in HAPLN^{high} tumors, and conclude that HAPLN is responsible for this observation. Does in vitro co-culture of HAPLN^{high} with CAFs induce the same iCAF phenotype and if so, can this phenotype be rescued by HAPLN knockdown and/or hyaluronidase? Can treating KPC cells with IL-6 or co-culture with iCAFs induce expression of HAPLN in a STAT3-dependent fashion?

5. Figure 5: How does HAPLN induce the systemic anti-inflammatory effects that the authors observe?

6. Figure 6: Is this process specific to peritoneal metastasis? What happens if HAPLN^{high} cells are injected into spleen (model of hematogenous/distant metastasis to liver) or tail vein (hematogenous to lungs), for example?

Minor points:

1. In the first sentence of the introduction (lines 43-45) the authors dogmatically state that PDAC metastasizes early, with no mention of which step(s) of the metastatic cascade the authors refer to. Although dissemination of single tumor cells can theoretically begin early even at the PanIN stage in mice (for example, Rhim et al. Cancer Cell 2012), it is unknown if early dissemination contributes to clinically relevant metastatic outgrowth. In fact, genetic evidence from patients indicates otherwise (for example, Yachida et al. Nature 2010). Because early vs. late metastasis holds major implications for the ability to detect and cure disease prior to clinically relevant

metastasis, please revise accordingly.

2. Further down in the introduction, (lines 71-78) the authors dogmatically state that tumor cells must proceed through an omental colonization intermediate step to disseminate to the rest of the peritoneal cavity. While omental "caking" is common in a subset of cancers (e.g. ovarian serous carcinomas), it is not an obligate step for peritoneal spread. In fact, peritoneal carcinomatosis by direct seeding from primary (in the absence of omental metastases) is not uncommonly observed including in patients with PDAC.

Reviewer #3 (Remarks to the Author):

The manuscript very nicely shows the relevance of HAPLN1 in pancreatic cancer patients. Using in vitro and in vivo models the authors demonstrate the role of HAPLN1 in migration, plasticity, extracellular matrix remodeling and immune suppression. This work might be relevant in the future to predict PDAC patients that will develop peritoneal carcinomatosis and maybe develop new therapies for these patients.

In my opinion the experiments done in this work support the conclusions claimed. However, some points need some clarification:

1. In figure 2F the authors use few genes to determine the stemness potential of KPC-HAPLN1 vs KPC cells. The text would benefit from a justification on why were these genes chosen out of all the stemness markers available.
2. The authors use flow cytometry to assess the percentage of live/dead cells and 3D organization (figure 2H). Flow cytometry does not conserve 3D structure and, therefore, 3D organization conclusions cannot be claimed from this experiment.
3. Some panels in figure 3 use area as measure of migration potential. However, the authors showed differences in cell shape and clustering (ext fig 2). Differences in migration potential would be more clear measuring total area by densely populated area (darker area).
4. In figure 3 authors show that HA/CD44 pathway drives cell migration changes in KPC-HAPLN1 vs KPC. This conclusion would be confirmed by showing that HA increase drives migration in KPC cells.
5. Figure 3D images do not show very accurately the results in the graphs in the same panel.

Also, some minor comments that would be interesting to complement the research are:

1. The authors made an experiment implanting KPC and KPC-HAPLN1 cells orthotopically in mice. In the first paragraph of the manuscript authors made a link between HAPLN1 expression and PDAC subtype. In the orthotopic models did the authors see differences regarding tumor subtype?
2. The manuscript would benefit from a description on how cells sit in the omentum in the peritoneal carcinomatosis model: differences and similarities in tumor size, morphology, are cells seeding less or more, migrating less or more.
3. In human datasets, is HAPLN1 differentially express in metastasis?
4. Line 356 have some spelling errors.

Reviewer #4 (Remarks to the Author):

In this manuscript, the authors studied an upregulated ECM protein HAPLN1 in PDAC and showed it functions through HA pathway to regulate cell plasticity, promote invasion and modulate immune cell atlas. The in vitro 3D culture system and the invasion-promoting function are quite well established. The authors also revealed cell state changes in tumor cells, one population of CAFs by high-throughput sequencing and also analyzed immune cell portions by FACs in PL in mice. There are some minor and major comments in various places.

Minor: In Figure 1F, the numbers of patients in each study and in each group (HAPLN1 high vs. low) need to be indicated. How were the patients divided into HAPLN1 high vs. low groups? It is missing from the methods.

Cellular origin question: HAPLN1 could be expressed from CAFs and cancer cells, and maybe other cells as well. Which type(s) of cells expresses HAPLN1 in PDAC? PDAC patient tissue IHC or sc-RNAseq public data analysis would give some info.

Non-cell-autonomous function confirmation: Secretion of HAPLN1 from KPC-HAPLN1 should be tested and compared to KPC cells. The pro-invasion phenotype should be confirmed with purified HAPLN1 protein or KPC- HAPLN1 condition media.

In vivo experiment-major issues:

The effect of HAPLN1 on tumor growth, metastasis and how immune modulation plays roles in tumor growth and/or metastasis need further investigation.

1. Better quantification of mets seems essential. Mice peritoneal metastases overview pictures should be shown. IP injection usually results in multifocal metastases. How was the tumor weight measured? Mouse luminescence images would help evaluate the tumor load.
2. The effect of HAPLN1 on tumor growth needs clarification. Does HAPLN1 impact on cancer cell proliferation and death in 2D and 3D culture?
3. There is lack of CD45+ cell infiltration into the peritoneal tumors, thus immune cell functions could not be analyzed in tumors, and instead, the authors turned to PL. It is difficult to link the immune cell profile in the fluid to the metastasis process in tumors. It seems when the tumors are well established, immune cells may not have pro-tumoral functions or are excluded by certain dominant inhibitory effects. What about only a few days after tumor cell injection? The immune cell profile could be surveyed and compared between KPC and KPC-HAPLN1 at an early stage of metastasis, such as colonization.
4. Does immune modulation impact on cancer cell proliferation and death? Cell proliferation and death could be evaluated in 3D cancer cell culture system and orthotopic mouse model which has microenvironment.

REVIEWER COMMENTS

Reviewer #1 (Remarks to the Author): Wiedmann et al. demonstrate HAPLN1 is enriched in the basal/mesenchymal PDAC and that functionally, expression of HAPLN1 leads to immunomodulation and increased peritoneal spread in a mouse model. Overall, peritoneal spread is a common path for PDAC metastasis that has significant effects on survival and a major source of morbidity. This study evaluates this significant problem and is important for the field. The work is done well with supportive human correlative data and preclinical models.

We thank the reviewer for the positive evaluation of our manuscript and the comments which helped to further improve our manuscript.

Comments/Questions

1) Introduction

a. The authors use EMP and EMT in the intro, but I would stick with EMT since this is what the field uses. EMP is certainly both EMT and MET so you could also just indicate EMT and MET have been shown and this represents plasticity.

We thank the reviewer for this helpful suggestion. We agree that using a more known term like EMT can reduce confusion and help understanding of the readers. We changed the paragraph accordingly: "Cellular plasticity is characterized by the ability of cells to convert between different (intermediate) cellular states by inducing epithelial-to-mesenchymal transition (EMT), as well as mesenchymal-to-epithelial transition (MET) and features of cancer cell stemness." (lines 62-65)

b. The section on omental metastasis followed by peritoneal metastasis makes it seem that this is the known path to the peritoneum, but I don't think this is known. I would rephrase to say that omental and peritoneal metastatic spread appears to be driven by a combination of niche and EMT

We agree that our phrasing overstates the state of knowledge in pancreatic cancer to this date. We corrected the indicated issue in the following way: "When metastasizing into the peritoneum, disseminated tumor cells can often colonize milky spots of the omental fat pad, a metastatic niche with high nutrient levels and immune cells with more resolving, anti-inflammatory features (mostly resident macrophages and B cells)." (lines 81-84)

2) Figure 1: Analysis of published expression profiles indicates HAPLN1 is associated with EMT and survival. This analysis is done well and I have no additional comments. One could also do a similar analysis on other data sets (TCGA, ICGC, etc.) to gain further confidence.

We again thank the reviewer for this suggestion, which has contributed to increase the relevance of our results. We have analyzed TCGA data available in the KM-plotter website and included these data in the suppl Figure 1E-G.

3) Figure 2: HAPLN1 overexpression in KPC cell lines shows higher HA, Has2, Vcan and EMT markers. Overall, experiments are reasonable. ? if there is any KPC line or growth conditions they have found with high endogenous HAPLN1 expression. This would probably be helpful to support physiological relevance of HAPLN1.

This is a very important question that we addressed intensively. However, we were unable to detect endogenous HAPLN1 expression in any of the pancreatic cancer cell lines we tested in vitro (KPC, KC, PANC-1, Bx-Pc3, CAPAN-1, PancO2). Thus, it seems that available cell lines do not express HAPLN1 endogenously. Nonetheless, following Reviewer #2's suggestion, we include now patient data (new Figure 1G, H) where we demonstrate the expression of HAPLN1 specifically in tumor cells in patients, supporting our approach. We think that HAPLN1 expression in cancer cell line depends on the tumor microenvironment. However, to further address this would be far beyond the scope of this manuscript.

4) Figure 3: I'm surprised by the inverse effect of MEKi on invasiveness. Most of the literature has indicated that KRAS->MEK signaling is a driver of EMT. Is this something unique to this cell line or this particular MEK inhibitor? Additional cell lines or other MEK inhibitors or CRISPR for MAPK would help determine this discrepancy with the broader literature.

Following Reviewer #1's suggestion, we repeated this experiment with a new cell line (PANC-1) and a new inhibitor (ERK Activation Inhibitor Peptide I). Again, we obtained the same effects of increased

invasion, when the ERK-MEK signaling cascade was blocked (**Supplementary Figure 5D**), indicating that – at least in our experimental models - ERK activation is a negative regulator of invasion.

In the revised manuscript we provide plenty of new data implicating TNF signaling through TNFR2 as a molecular mechanism explaining the effect of HAPLN1-induced invasion. Since TNF α acts through different receptors and they act through different signaling pathways (please see below), we hypothesize that blocking ERK activation, which takes place upon TNFR1, could increase the signaling through TNFR2. This has been added to the Discussion.

[REDACTED]

5) Figure 4: In vivo data showing changes in PDAC and CAF cells. RNA-seq in PDAC cells shows loss of Acan1 c/w in vitro and gain of EMT genes. Overall the data is done well but consider putting KPC vs KPC-HAPLN1 directly on the volcano plots to improve clarity.

Thank you very much for pointing out our omission. We changed the Figure in the proposed way (New Figure 4B).

6) Figure 5/6: Immune and metastasis in animal data. Overall really well done. ? if the authors can expand in the discussion thoughts on the relative contribution of HAPLN1 metastasis to immune/Macrophage changes vs EMT changes?

We thank the reviewer for this important remark. We have expanded this subject in the Discussion and we strongly believe that it is now satisfactory.

Reviewer #2 (Remarks to the Author):

In this work the authors explored the contribution of HAPLN to peritoneal metastasis by pancreatic ductal adenocarcinoma (PDAC). Because HAPLN is known to be important in other tumor contexts, expression was evaluated in publicly available datasets which showed increased HAPLN in PDAC primary tumors (vs normal) and shortened overall survival times. To investigate HAPLN experimentally, the authors (over)-expressed HAPLN in a KPC cell line (HAPLN^{high}). Over-expressing HAPLN resulted in gene expression and spheroid changes suggestive of increased HA production, cell plasticity, and invasion. i.p. injection of tumor cells showed that CAFs and the immune landscape in tumors formed by HAPLN^{high} diverged from HAPLN^{low}, and that HAPLN^{high} cells more efficiently colonized the peritoneal cavity than HAPLN^{low} cells.

Although intriguing and potentially of high importance, the data are descriptive, lack appropriate functional interrogation, and no mechanistic insights are provided for how HAPLN might cause the observed changes or promote peritoneal-specific metastasis (as opposed to distant/hematogenous

metastasis or primary tumor growth). In my opinion the work as it stands is preliminary would require additional experimentation to achieve suitable rigor and mechanistic insights, as outlined below.

We thank the reviewer for the positive evaluation of our manuscript and for pointing out the relevance of our work. We appreciate the comments which helped to further improve our manuscript. In particular we have now added a series of new data showing more and deeper mechanistic insights.

General Points:

1. Figure 1: The authors report that HAPLN^{high} is enriched in public data of patient primary tumors, including subtypes with a basal-like signature. However, it is unclear if expression is from tumor cells themselves, from stroma constituents (e.g. CAFs), or both. Are these samples bulk tumor tissues or microdissected? This is especially pertinent considering primary PDACs are often very stroma rich, and other studies have shown HAPLN is secreted into stroma by CAFs.

The patient data used are derived from bulk tumors. However, we have addressed this point in two steps. Firstly, we have analyzed patient data of cell type specific RNA profiles. We observed that HAPLN1 mRNA expression increases only in epithelial cells although being detected also in other cell types (New Figure 1G).

Secondly, we stained patient samples for HAPLN1 by IHC and observed that HAPLN1 protein is indeed expressed in tumor cells (New Figure H). [The analysis was done by board-certified pathologists H.B. and P.S.]

However, given that CAFs also expressed high amounts of HAPLN1, we considered the possibility that our samples had a higher content of CAFs, which are well known to contribute to worse prognosis in patients and could be responsible for the mesenchymal profile observed. To rule this out, we analyzed the clinical data included in the article in which this dataset was published (Cao et al., Cell 2021) and evaluated the cellular component of the analyzed samples. Interestingly, the stromal component was not different between the two groups of patients, while the epithelial content was significantly higher in HAPLN1^{high} patients compared to HAPLN1^{low} (new Supplementary Figure 1H).

This shows that the increased HAPLN1 expression was not attributable to an increased CAF content in those samples.

2. Figures 2-6: All experimental data is based on a KPC cell line with forced over-expression of HAPLN. This is a very limited reagent set (one cell line) using a very limited approach (over-expression). KPC mice themselves are well-known to spontaneously develop peritoneal metastasis and malignant ascites fluid (from primary tumors). The work would be strengthened by examination of HAPLN in the natural context of peritoneal metastasis complemented by loss-of-function approaches from cells isolated from these or similar such lesions.

We thank the reviewer for this remark, because we agree that using a single cell line for all the experiments is not ideal. We are aware of the literature showing that also in KPC mice the spontaneous tumors metastasize into the peritoneum, however this happens at very late stages, which we cannot reach due to the European animal welfare regulations. Importantly, we never wanted to claim that KPC cells were not able to metastasize into the peritoneal cavity. The message that we wanted to convey is that the expression of HAPLN1 accelerates the metastatic spread to the peritoneal cavity. In other words, we do not claim that HAPLN1 is essential for peritoneal colonization, but rather that it potentiates the features necessary for this colonization. In an attempt to increase the relevance of our results, we employed another, this time human, pancreatic cancer cell line called PANC1 to reproduce the most crucial data obtained with the KPC tumor cell line. Also here, we could see an improved spheroid formation (new Suppl. Figure 2G)

and an increase in their invasive ability (new Figure 3C) upon HAPLN1 expression.

In addition, in an attempt to better understand the mechanism by which HAPLN1 potentiates plasticity, we observed that TNF α appeared as the estimated upstream regulator in the unbiased analysis of the *in vivo* RNAseq data obtained from tumor cells. This prompted us to analyze the contribution of TNF α to the process, which is explained later in this document. We want to highlight here that we also found that TNF α increased invasion in PANC1, not only in KPC (see answer to point 3) and that it mediates HAPLN1-induced invasion (new Supplementary Figure 5B).

Lastly, we would like to mention that we fully agree that loss-of-function experiments would be interesting. However, we were unable to detect endogenous HAPLN1 expression in any of the pancreatic cancer cell lines we tested *in vitro* (KPC, KC, PANC-1, Bx-Pc3, CAPAN-1, PancO2). We think that HAPLN1 expression in cancer cell line depends on the tumor microenvironment. However, to further address this would be far beyond the scope of this manuscript.

3. Figures 2-6: It seems the authors assume that HAPLN must mediate all the observed effects through linking cells to HA and/or stimulating HA synthesis/production. However, this is never addressed mechanistically or tested functionally. It is unclear how one protein or a single protein:matrix interaction would mediate all the observed effects. This is especially relevant considering that HAPLN can be either a tumor suppressor or oncogene depending on the context, and PEGPH20 has not fared well in clinical trials for PDAC patients.

Based on your suggestion, we tried to better understand the effect of HAPLN1 on tumor cells by addressing two major points:

- 1) Better understanding of the effect of recombinant HAPLN1 and HA on tumor cell invasion
- 2) Investigating for an upstream regulator responsible for the seen effects on HA and tumor cell behavior.

To address 1), we performed experiments of invasion into a collagen matrix which, unlike Matrigel, is a very basic matrix without unknown factors that might influence cellular behavior. In this setting we tested the impact of HA and recombinant HAPLN1 on KPC cells, since these might be already present in the Matrigel composition, thereby affecting our results. We detected an increase in invasion with recombinant HAPLN1, although it did not reach statistical significance. Moreover, we observed significant increase of invasion when treating KPC spheroids with HA or HA and recombinant HAPLN1.

Additionally, we tried to block the effect of HA in several different ways. While using Hyaluronidase (HAase), which processes polymers of HA into oligomers but does not eliminate HA, increased the invasiveness of the cells, while blocking HA production by 4-MU completely ablated invasion, suggesting that HA, regardless of its size is crucial for invasion. On the other hand, blocking CD44 with an antibody did not affect it significantly, suggesting a CD44 independent response.

Taking both results together, HA production is essential for cell invasion, HAPLN1 potentiates invasion two-fold, by inducing HA production and by interacting with HA.

To address point 2), we made use of our RNAseq data obtained from the sorted tumor cells extracted from the solid tumors formed and used Ingenuity Pathway Analysis (IPA) to search for upstream regulators. The upstream regulator most significantly upregulated in KPC-HAPLN1 cells compared to KPC cells was TNF α (new Figure 5E).

Intriguingly, we found TNF α signaling as well as TNFR2 expression significantly increased in KPC-HAPLN1 cells in vitro and in vivo, strengthening the data obtained by IPA (new Figures 4G and 5A).

Additionally, we found that stimulating KPC and KPC-HAPLN1 cells with TNF α led to increased Has2 expression only in KPC-HAPLN1 cells, thereby strongly potentiating HA production. Moreover, CD44 was not differentially regulated, indicating that its TNF-mediated upregulation is independent of TNFR2 signaling and confirming the CD44-independent effect observed before (new Figure 5G).

We also observed that TNF α induced a similar level of invasion in KPC and PANC1 cells as HAPLN1 overexpression did, while blocking TNF α reverted HAPLN1-induced invasion (new Figure 5D,E and new Suppl. Figure 5B).

This effect could only partially be rescued by a CD44 blocking antibody, but fully when blocking HA synthesis by 4-MU, suggesting that the effect is mediated by HA. (Figure 5F).

We want to thank the reviewer very much for pointing out a certain lack of mechanistic insights. We believe that based on this suggestion, we have improved our manuscript substantially.

Technical Points:

1. Figure 1F: The survival curves for mRNA and protein seem strikingly similar. Please provide more clarity on the methodology behind these plots, especially how the cut-off between HAPLN^{high} and HAPLN^{low} based on mean values was determined. Was the protein data extracted from pathology scoring (by IHC), digital quantification, mass spectrometry, or some other method? I could not find any of this information in the legend or the methods.

Please excuse us for our omission. We included a new paragraph in the M&M section describing how we performed the classification. Here the paragraph added to the manuscript:

“Classification in HAPLN^{high} and HAPLN^{low} cohorts for survival, GSEA and cellular content

For the survival data we made use of the publicly available transcriptomic, proteomic and clinical data published in Cao et al. 2021 24. For the distribution of patients into the HAPLN^{high} or HAPLN^{low} cohorts, we extracted the HAPLN1 expression (obtained by RNA-sequencing) or HAPLN1 protein levels (obtained by mass spectrometry) and calculated the mean expression levels over all patients. Patients were assigned to the HAPLN^{high} cohort, if their HAPLN1 expression level was above this mean, and

vice versa. mRNA and protein analyses were conducted independently. Then we ordered patients according to follow up days and vital status, and plotted resulting survival curves. Same classification was employed for the analysis of the cellular content of the tumors, using the clinical data provided by the authors of the dataset.” (lines 670-681)

2. Figure 2C: The authors show that KPC cells are HAPLN^{low}, necessitating forced over-expression of HAPLN. However, as pointed out above KPC mice develop peritoneal metastasis and the authors also show high HAPLN expression in patient primary tumors. Can the authors address these discrepancies?

We did not intend to state that KPC cells necessitate HAPLN1 over-expression to become metastatic. What we show with our data is that higher concentrations of HAPLN1 accelerates the process. Those patients whose HAPLN1 expression is higher (compared to the mean of all the patients) will have worse prognosis, probably due to an earlier development of peritoneal carcinomatosis. This does not imply that patients whose HAPLN1 expression is lower will not develop similar complications eventually. Indeed, we found alive tumor cells in the peritoneal lavage also in KPC injected mice, but the number was so small, that it would be necessary to have a much more advanced tumor size to achieve the same number as with KPC-HAPLN1 injected. Unfortunately, such tumor size does not comply with animal welfare regulations in Europe.

Most of the transcriptomic data available have been performed in resectable tumors. These are often early stages of tumor development. Indeed, in the new patient data provided (new Figure 1G) we observed that only some of the patients increase HAPLN1 expression in their tumor cells. It is very possible that only some of those tumor cells will express HAPLN1, because we know that tumors are very heterogenous. When addressing this heterogeneity in the laboratory, we opted for comparing the two extremes to reveal the consequences of HAPLN1 presence in the microenvironment. Given that in our culture conditions, all the cell lines evaluated showed a low expression of HAPLN1, we opted to over-express it.

3. Figure 3: Rescue of inhibitor effects (with exogenous HA for example) and phenocopy of inhibitor effects (with CD44 knockdown, HAPLN knockdown, hyaluronidase Tx's, as examples) would greatly strengthen the confidence that HA/CD44/HAPLN mediate the observed spikey (“invasion”) morphology observed in matrigel.

Thank you very much for this suggestion. As already pointed out above, we did perform invasion experiments using exogenous HA, recombinant HAPLN1, CD44 inhibition, hyaluronidase treatment and inhibition of HA production by 4-MU to address these points. We did not use HAPLN1 knockdown, since we were not able to find a cell line expressing HAPLN1 in vitro, which made a knockdown irrelevant. We found that HAPLN1, promotes TNFR2 expression to increase TNF α -induced HAS2 expression. HA production is essential for invasion, but in a CD44-independent manner.

4. Figure 4: The authors detect enrichment for inflammatory-type CAFs (“iCAFs”) in HAPLN^{high} tumors, and conclude that HAPLN is responsible for this observation. Does in vitro co-culture of HAPLN^{high} with CAFs induce the same iCAF phenotype and if so, can this phenotype be rescued by HAPLN knockdown and/or hyaluronidase? Can treating KPC cells with IL-6 or co-culture with iCAFs induce expression of HAPLN in a STAT3-dependent fashion?

Thank you very much for this interesting suggestion. However, we first of all would like to state that we did not claim an enrichment of inflammatory CAFs, since we lack the proper investigation to draw these kind of assumptions. The CAFs we analyzed by RNAseq were bulk, which made it impossible to understand if there had been a shift in CAF subpopulations. We only wanted to demonstrate that the

CAF transcriptome had significant alterations, which seemed to be mainly focused on e.g. an upregulation of cytokine production in the KPC HAPLN1 tumors. Although very interesting, investigating this further was out of the scope of this manuscript.

Since your suggestions anyways caught our attention, we cultured GRX fibroblasts (activated hepatic stellate cell line; hepatic stellate cells are quite similar to pancreatic stellate cells, and for this reason we deemed appropriate to use this cell line) on top of a conditioned matrix derived from KPC or KPC HAPLN1 tumor cells, to understand if the HAPLN1 in the ECM did have an impact on the fibroblasts. Indeed, we could detect a significant upregulation of the inflammatory CAF marker *Lif* in GRX cultured on the conditioned matrix of KPC HAPLN1 cells, while *Col1a2* was significantly reduced.

However other markers, like *Il6*, were unchanged, suggesting that not all the changes detected in vivo could be reproduced with this experiment in vitro. We did not perform a HAPLN1 knockdown, since the control condition has low *Hapln1* expression, and we could not find a cell line that expressed *Hapln1* in vitro to do knockdown studies.

- Figure 5: How does HAPLN induce the systemic anti-inflammatory effects that the authors observe? - Thanks to the reviewers' suggestions we have now performed several new experiments and the new data point out to a contribution of increased expression of *TNFR2* as a mechanism by which HAPLN1 increases invasion. Although this is not a primary focus of our paper, we also observed a more tolerogenic environment in the peritoneal cavity. Interestingly, it has been previously described a role for *TNFR2* in immunosuppression in colorectal cancer, in particular in the metastatic process towards the liver (PMID: 26483205). In this study, authors related *TNFR2* expression with increases in Tregs and MDSCs, generating a more tolerogenic microenvironment in the liver. Moreover, a recent meta-analysis concluded that soluble *TNFR2* is a potential biomarker for cancer diagnosis (PMID: 36466914), supporting the role of this receptor systemically. We can speculate that a similar mechanism is taking place in our model. Moreover, the combined expression of *TNFR2* and *MCH-II* observed in KPC-HAPLN1 cells in vivo, also suggests a potential role of those particular tumor cells on immunosuppression. However, the primary focus of our study is the analysis of HAPLN1-induced migration, for that we believe that the mechanisms regulating systemic inflammation are beyond its scope.
- Figure 6: Is this process specific to peritoneal metastasis? What happens if HAPLNhigh cells are injected into spleen (model of hematogenous/distant metastasis to liver) or tail vein (hematogenous to lungs), for example? This is a very relevant question, and it would be very interesting to investigate it. However, we believe that it is out of the scope of this manuscript. In this study we specifically aimed to identify a novel marker for accelerated peritoneal metastasis. We did not want to focus on the liver metastasis route, and we do not want to make any assumption on that, because we do not possess relevant data to speculate on that. Nonetheless, we will probably address this on future studies.

Minor points:

1. In the first sentence of the introduction (lines 43-45) the authors dogmatically state that PDAC metastasizes early, with no mention of which step(s) of the metastatic cascade the authors refer to. Although dissemination of single tumor cells can theoretically begin early even at the PanIN stage in mice (for example, Rhim et al. Cancer Cell 2012), it is unknown if early dissemination contributes to clinically relevant metastatic outgrowth. In fact, genetic evidence from patients indicates otherwise (for example, Yachida et al. Nature 2010). Because early vs. late metastasis holds major implications for the ability to detect and cure disease prior to clinically relevant metastasis, please revise accordingly.

Thank you for correcting this important point of our overstatement. We changed the mentioned sentence accordingly.

“Pancreatic ductal adenocarcinoma (PDAC) ranks among the most lethal cancer entities, with late diagnosis as key contributor to its poor survival rate, as most patients are detected with metastatic disease.” (lines 50-52)

2. Further down in the introduction, (lines 71-78) the authors dogmatically state that tumor cells must proceed through an omental colonization intermediate step to disseminate to the rest of the peritoneal cavity. While omental “caking” is common in a subset of cancers (e.g. ovarian serous carcinomas), it is not an obligate step for peritoneal spread. In fact, peritoneal carcinomatosis by direct seeding from primary (in the absence of omental metastases) is not uncommonly observed including in patients with PDAC.

We have re-stated the mentioned lines as follows:

“When metastasizing into the peritoneum, disseminated tumor cells can often colonize milky spots of the omental fat pad, a metastatic niche with high nutrient levels and immune cells with more resolving, anti-inflammatory features (mostly resident macrophages and B cells) ¹⁰. For instance, omental resident macrophages were proven crucial for metastatic progression and immunomodulation ¹¹. Thus, peritoneal and omental metastasis are driven by a combination of tumor cell intrinsic plasticity and niche.

When first colonizing the omentum, cancer cells progress to spread throughout the whole peritoneum.” (lines 81-89)

Reviewer #3 (Remarks to the Author):

The manuscript very nicely shows the relevance of HAPLN1 in pancreatic cancer patients. Using in vitro and in vivo models the authors demonstrate the role of HAPLN1 in migration, plasticity, extracellular matrix remodeling and immune suppression. This work might be relevant in the future to predict PDAC patients that will develop peritoneal carcinomatosis and maybe develop new therapies for these patients.

In my opinion the experiments done in this work support the conclusions claimed. However, some points need some clarification:

We thank the reviewer for the positive evaluation of our manuscript and the constructive comments.

1. In figure 2F the authors use few genes to determine the stemness potential of KPC-HAPLN1 vs KPC cells. The text would benefit from a justification on why were these genes chosen out of all the stemness markers available.

We designed these experiments as a proof of concept. For this reason, we analyzed the expression of a few genes that literature commonly associates with stemness. For instance, ABC transporters have been shown to be commonly expressed in cancer stem cells (CSCs) (PMID: 15803154) and PLOD2 was recently described as a marker for CSCs in PDAC (PMID: 32958051) We have included these citations in the manuscript to support our panel.

We have now included some more commonly known CSC genes (c-kit, CD133, Lgr5) in our panel (which were also upregulated in the RNAseq data) to increase the selection (new Suppl. Figure 2F).

2. The authors use flow cytometry to assess the percentage of live/dead cells and 3D organization (figure 2H). Flow cytometry does not conserve 3D structure and, therefore, 3D organization conclusions cannot be claimed from this experiment.

We apologize for not explaining this in a better way. We did not want to claim anything about the 3D organization by using flow cytometry. We speculated that the increased viability in KPC-HAPLN1 cells, which was analyzed by flow cytometry after disrupting the spheroids, could be attributed to a better 3D organization of their organoids. In any case, to avoid confusion, we have eliminated this comment from the manuscript.

3. Some panels in figure 3 use area as measure of migration potential. However, the authors showed differences in cell shape and clustering (ext fig 2). Differences in migration potential would be more clear measuring total area by densely populated area (darker area).

We thank the reviewer for this suggestion and agree that it would be a better way to measure the invaded area. For this we first planned to normalize data to the area occupied by the spheroids before the addition of Matrigel as a measure for the non-invaded area. However, due to the change from aggregates to spheroids in KPC by the addition of Matrigel, these spheres seemed to be shrinking, which made the data impossible to compare to the relatively similar levels of KPC HAPLN1 spheroids.

For this reason, we appreciated your suggestion. However, when trying to perform this analysis, we realized that defining the border of the dense and the invaded area was not automatic and seemed a bit arbitrary. Below we would like to give you some examples to explain this better and why we decided to leave the analysis as it was before:

Here we show two examples where the upper panel shows less migration than the lower. However, when applying the same threshold to determine the dense area, the area selected does not correspond to the intended one. This could be solved by drawing the area, however, we do not think that this would be objective enough to qualify as a proper quantification. Therefore, we have decided to leave the quantification as originally shown.

4. In figure 3 authors show that HA/CD44 pathway drives cell migration changes in KPC-HAPLN1 vs KPC. This conclusion would be confirmed by showing that HA increase drives migration in KPC cells.

This question was similar to a point raised by Reviewer #2. Indeed, addressing to their question we were able to spot a misinterpretation of our results. We found that HA-mediated increase in invasion is independent of CD44. This mistake stems from the fact that CD44 shares very similar signaling to TNFR2, which we know now that is mediating most of the effects for HAPLN1-induced invasion. We have modified the text accordingly. We reproduce here our answer to Reviewer #2 because we think that it is relevant to answer this question: we performed experiments of invasion into a collagen matrix which, unlike Matrigel, is a very basic matrix without unknown factors that might influence cellular behavior. In this setting we tested the impact of HA and recombinant HAPLN1 on KPC cells, since these might be already present in the Matrigel composition, thereby affecting our results. We detected an increase in invasion with recombinant HAPLN1, although it did not reach statistical significance. Moreover, we observed significant increase of invasion when treating KPC spheroids with HA or HA and recombinant HAPLN1.

Additionally, we tried to block the effect of HA in several different ways. While using Hyaluronidase (HAase), which processes polymers of HA into oligomers but does not eliminate HA, increased the invasiveness of the cells, while blocking HA production by 4-MU completely ablated invasion,

suggesting that HA, regardless of its size is crucial for invasion. On the other hand, blocking CD44 with an antibody did not affect it significantly, suggesting a CD44 independent response.

Taking both results together, HA production is essential for cell invasion, HAPLN1 potentiates invasion two-fold, by inducing HA production and by interacting with HA.

5. Figure 3D images do not show very accurately the results in the graphs in the same panel.

Following reviewer's suggestions, we have selected different images that more accurately reflect the mean values in the quantification (new Figure 3E). We hope that they are more convincing.

Also, some minor comments that would be interesting to complement the research are:

1. The authors made an experiment implanting KPC and KPC-HAPLN1 cells orthotopically in mice. In the first paragraph of the manuscript authors made a link between HAPLN1 expression and PDAC subtype. In the orthotopic models did the authors see differences regarding tumor subtype?

This is a very interesting question, which we would have liked that it raised earlier in our project. The primary focus of this paper was to analyze invasion. When we performed the initial experiment orthotopically and saw not effects on invasion at the time points that regulation allowed us to evaluate, we decided to cancel these experiments. Injection of tumors in the pancreas is a very variable procedure, which often resulted in very different primary tumors. One of the few shared features among tumors within the same group was their size. But variance in shape was too high to be able to conclude anything about their subtype with the few samples that we had obtained (n=6).

However, since we find this suggestion very important, we asked a pathologist to analyze the solid tumors formed in the omentum of the mice bearing tumors injected intraperitoneally, because the variation within each group was very small. Interestingly, nodules formed by KPC-HAPLN1 cells did not show any glandular structures (a sign of a more differentiated tumor subtype), which goes along with the basal subtype, because this subtype is known to be less differentiated.

However, since this this is not a primary tumor, we thought that the inclusion of these data into the manuscript could be a bit confusing. Therefore, we have not included it in the revised manuscript. We hope that it will in any case, convince the reviewer that our data is relevant.

2. The manuscript would benefit from a description on how cells sit in the omentum in the peritoneal carcinomatosis model: differences and similarities in tumor size, morphology, are cells seeding less or more, migrating less or more.

We already outlined some of these differences in the manuscript, and following reviewers' suggestions we have expanded them. In the revised version we provide new data regarding in vivo luminescence of our tumors. We found increased luminescence in KPC-HAPLN1 injected mice already at 8 d.p.i. (new Figure 4A).

Considering that tumor weight is lower at our end point (11 d.p.i.), this indicates that the increased luminescence was derived from cells present in the peritoneal cavity. Morphologically, we could detect differences in the formation of ductal structures, as pointed out before. Also macroscopically, tumors were very different (see image below, middle panel). However, we do not know how to interpret these differences. We are certain that the molecular and transcriptomic differences obtained are behind this very different tumor shapes, but this will probably be part of a follow up study.

3. In human datasets, is HAPLN1 differentially express in metastasis?

We agree that this would be a very nice way to validate our findings. However, we are not aware of any human PDAC dataset in which cancer cells from peritoneal metastasis were analyzed, which makes it impossible for us to perform this kind of analysis. Nevertheless, we want to thank the reviewer for this important suggestion.

4. Line 356 have some spelling errors.

Thank you very much for pointing out the errors. We corrected them.

Reviewer #4 (Remarks to the Author):

In this manuscript, the authors studied an upregulated ECM protein HAPLN1 in PDAC and showed its functions through HA pathway to regulate cell plasticity, promote invasion and modulate immune cell atlas. The in vitro 3D culture system and the invasion-promoting function are quite well established. The authors also revealed cell state changes in tumor cells, one population of CAFs by high-throughput sequencing and also analyzed immune cell portions by FACs in PL in mice. There are some minor and major comments in various places.

We thank the reviewer for the positive evaluation of our manuscript and the useful comments.

Minor: In Figure 1F, the numbers of patients in each study and in each group (HAPLN1 high vs. low) need to be indicated. How were the patients divided into HAPLN1 high vs. low groups? It is missing from the methods.

We added the group sizes in the Figures. Also, we want to apologize for forgetting to add the description into the methods part. We updated it accordingly:

“Classification in HAPLN1high and HAPLN1low cohorts for survival, GSEA and cellular content

For the survival data we made use of the publicly available transcriptomic, proteomic and clinical data published in Cao et al. 2021 24. For the distribution of patients into the HAPLN1high or HAPLN1low cohorts, we extracted the HAPLN1 expression (obtained by RNA-sequencing) or HAPLN1 protein levels (obtained by mass spectrometry) and calculated the mean expression levels over all patients. Patients were assigned to the HAPLN1high cohort, if their HAPLN1 expression level was above this mean, and vice versa. mRNA and protein analyses were conducted independently. Then we ordered patients according to follow up days and vital status, and plotted resulting survival curves. Same classification was employed for the analysis of the cellular content of the tumors, using the clinical data provided by the authors of the dataset.” (lines 670-681).

Cellular origin question: HAPLN1 could be expressed from CAFs and cancer cells, and maybe other cells as well. Which type(s) of cells expresses HAPLN1 in PDAC? PDAC patient tissue IHC or sc-RNAseq public data analysis would give some info.

We thank the reviewer for this important comment. We reproduce here the answer to reviewer #2 who also addressed this point:

We have addressed this point in two steps. Firstly, we have analyzed patient data of cell type specific RNA profiles. We observed that HAPLN1 expression increases only in epithelial cells. Also, we stained patient samples for HAPLN1 by IHC and observed that HAPLN1 is indeed expressed in tumor cells (New Figure 1G, H).

However, given that CAFs also expressed high amounts of HAPLN1, we considered the possibility that our samples had a higher content of CAFs, which are well known to contribute to worse prognosis in patients and could be responsible for the mesenchymal profile observed. To rule this out, we analyzed the clinical data included in the article in which this dataset was published (Cao et al., Cell 2021) and evaluated the cellular component of the analyzed samples. Interestingly, the stromal component was not different between the two groups of patients, while the epithelial content was significantly higher in HAPLN1^{high} patients compared to HAPLN1^{low} (new Supplementary Figure 1H).

This shows that the increased HAPLN1 expression was not attributable to an increased CAF content in those samples.

Non-cell-autonomous function confirmation: Secretion of HAPLN1 from KPC-HAPLN1 should be tested and compared to KPC cells. The pro-invasion phenotype should be confirmed with purified HAPLN1 protein or KPC- HAPLN1 condition media.

We tried to detect secreted HAPLN1, however it is very difficult to detect the presence of an ECM-linked protein outside of a cell, since it is not present in the supernatant and stainings by immunocytochemistry are difficult to interpret, given that there is no specific location within e.g. a cellular compartment. Thus, we were unable to demonstrate the secretion of HAPLN1. Nonetheless, following reviewers' suggestions, we have performed experiments using recombinant HAPN1 protein and exogenous HA. These experiments were performed in collagen, instead of Matrigel, since HA might be already present in the Matrigel composition, thereby affecting our results. We detected an increase in invasion with recombinant HAPLN1, although it did not reach statistical significance. Moreover, we observed significant increase of invasion when treating KPC spheroids with HA or HA and recombinant HAPLN1 (new Figure 3G).

We think that these results confirm the non-cell-autonomous effect of HAPLN1.

In vivo experiment-major issues:

The effect of HAPLN1 on tumor growth, metastasis and how immune modulation plays roles in tumor growth and/or metastasis need further investigation.

1. Better quantification of mets seems essential. Mice peritoneal metastases overview pictures should be shown. IP injection usually results in multifocal metastases. How was the tumor weight measured? Mouse luminescence images would help evaluate the tumor load.

Thank you for this comment. We measured the tumor weight only focusing on the big tumor nodules formed in the omentum. Other than that, we could not identify nodules by eye. We thank the reviewer for the suggestion to use in vivo luminescence to detect all tumor cells present in the peritoneal cavity. Like this, we detected significantly more in vivo luminescence in KPC HAPLN1 tumor bearing mice (new Figure 4A), even though their solid tumor weight was significantly less (new Suppl. Figure 4A), reinforcing our hypothesis of a more spread, detached tumor growth in mice injected with KPC-HAPLN1 cells.

2. The effect of HAPLN1 on tumor growth needs clarification. Does HAPLN1 impact on cancer cell proliferation and death in 2D and 3D culture?

Once more, we thank the reviewers for their suggestions, because we agree that addressing this more thoroughly was important. In the previous version of our manuscript we evaluated viability in 3D (new Figure 2H).

In the previous version we included also a doubling time experiment in 2D that showed no differences between KPC and KPC-HAPLN1 cells (new Suppl. Figure 2D).

We have performed a cell cycle analysis of the cells and found that in 2D culture, KPC-HAPLN1 cells do not proliferate more than KPC, in fact they proliferate less (new Suppl. Figure 2E).

We believe that these are convincing enough evidences to show that KPC-HAPLN1 effects are not attributable to an increased proliferation.

3. There is lack of CD45+ cell infiltration into the peritoneal tumors, thus immune cell functions could not be analyzed in tumors, and instead, the authors turned to PL. It is difficult to link the immune cell profile in the fluid to the metastasis process in tumors. It seems when the tumors are well established, immune cells may not have pro-tumoral functions or are excluded by certain dominant inhibitory effects. What about only a few days after tumor cell injection? The immune cell profile could be surveyed and compared between KPC and KPC-HAPLN1 at an early stage of metastasis, such as colonization.

The rationale behind studying the peritoneal lavage (PL) is that it can be considered as the tumor microenvironment in this setting. Cells have been injected into the peritoneum, and these formed the solid masses as mentioned before. So, the immune cell exclusion in them is an active event happening as part of tumor progression. Our new luminescence data show differences already 8 d.p.i. suggesting that this is an early event in tumor progression. Monitoring immune cell infiltration at even earlier time-points is very interesting, and it will probably be part of a future study together with the mechanism behind systemic effects on inflammation. However, our study is focused on tumor cell invasion, and for this reason we believe that studying the tumor cell presence in the PL was the main result.

4. Does immune modulation impact on cancer cell proliferation and death? Cell proliferation and death could be evaluated in 3D cancer cell culture system and orthotopic mouse model which has microenvironment.

We analyzed tumor masses for proliferation features and detected no significant differences (analysis performed by board certified pathologists H.B. and C.M), which discouraged us to evaluate this in any other setting. We have not included these data in the manuscript, but we show it here for the reviewers:

In order to answer the role of immune cells in cell death, we analyzed CD45⁺/RFP⁺ cell populations in the PL. We have excluded this population from our analysis because we were looking for tumor cells (CD45⁻/RFP⁺). However, we realized that this population is a measure of tumor cells phagocytosed by immune cells. We observed that immune cells were more efficient at phagocytosing tumor cells in KPC-injected mice (new Suppl. Figure 7B).

This highlights the major role of peritoneal immune cells on the regulation of peritoneal carcinomatosis.

REVIEWERS' COMMENTS

Reviewer #1 (Remarks to the Author):

The authors have addressed my main concerns/questions.

Reviewer #2 (Remarks to the Author):

Although I still feel a bit queezy about results based largely on a CMV over-expressing cell line, on the whole the authors have made significant efforts to address my points with new and convincing data, including mechanistic studies that have improved the work.

My only additional comment concerns the title, which by the authors own admission is misleading. The authors state in their rebuttal:

"we do not claim that HAPLN1 is essential for peritoneal colonization, but rather that it potentiates the features necessary for this colonization"

"Driver" is a grossly over-used term that should be limited to phenomenon that are clearly recurrent across patients (no data on spontaneous peritoneal mets in patients or animals is shown) and is required for the process by increasing fitness. It also typically refers to genetically encoded events.

I would suggest revising the title to better reflect the data, in that "HAPLN1 potentiates pancreatic cancer peritoneal metastasis by stimulating TNF α -dependent hyaluronic acid production"...or something similar.

Reviewer #3 (Remarks to the Author):

The authors have improved significantly the manuscript and addressed most of my concerns. I have only a minor issue remaining, in lines 170-183 Suppl. figure 2 panels are mislabeled. I don't see any reason to delay further the publication of these interesting results.

Reviewer #4 (Remarks to the Author):

The revision addressed most of my concerns. Together with the experiments done for other reviewers, I think the quality grants its publishing in NC.

Reviewer #1 (Remarks to the Author):

The authors have addressed my main concerns/questions.

Thank you very much.

Reviewer #2 (Remarks to the Author):

Although I still feel a bit queezy about results based largely on a CMV over-expressing cell line, on the whole the authors have made significant efforts to address my points with new and convincing data, including mechanistic studies that have improved the work.

My only additional comment concerns the title, which by the authors own admission is misleading. The authors state in their rebuttal:

"we do not claim that HAPLN1 is essential for peritoneal colonization, but rather that it potentiates the features necessary for this colonization"

"Driver" is a grossly over-used term that should be limited to phenomenon that are clearly recurrent across patients (no data on spontaneous peritoneal mets in patients or animals is shown) and is required for the process by increasing fitness. It also typically refers to genetically encoded events.

I would suggest revising the title to better reflect the data, in that "HAPLN1 potentiates pancreatic cancer peritoneal metastasis by stimulating TNFa-dependent hyaluronic acid production"...or something similar.

Thank you very much. We agree that driver might be a bit overstating the role of HAPLN1, for this we suggest: "HAPLN1 potentiates peritoneal metastasis in pancreatic cancer" as the new title.

Reviewer #3 (Remarks to the Author):

The authors have improved significantly the manuscript and addressed most of my concerns. I have only a minor issue remaining, in lines 170-183 Suppl. figure 2 panels are mislabeled. I don't see any reason to delay further the publication of these interesting results.

Thank you very much. We have corrected these mistakes

Reviewer #4 (Remarks to the Author):

The revision addressed most of my concerns. Together with the experiments done for other reviewers, I think the quality grants its publishing in NC.

Thank you very much.